# Economic damages from Hurricane Sandy attributable to sea level rise caused by anthropogenic climate change

Benjamin H. Strauss [1✉], Philip M. Orton[2], Klaus Bittermann [3,4], Maya K. Buchanan [1], Daniel M. Gilford [1,5], Robert E. Kopp [5], Scott Kulp[1], Chris Massey [6], Hans de Moel [7] & Sergey Vinogradov[2,8]

In 2012, Hurricane Sandy hit the East Coast of the United States, creating widespread coastal flooding and over $60 billion in reported economic damage. The potential influence of climate change on the storm itself has been debated, but sea level rise driven by anthropogenic climate change more clearly contributed to damages. To quantify this effect, here we simulate water levels and damage both as they occurred and as they would have occurred across a range of lower sea levels corresponding to different estimates of attributable sea level rise. We find that approximately $8.1B ($4.7B–$14.0B, 5th–95th percentiles) of Sandy's damages are attributable to climate-mediated anthropogenic sea level rise, as is extension of the flood area to affect 71 (40–131) thousand additional people. The same general approach demonstrated here may be applied to impact assessments for other past and future coastal storms.

[1] Climate Central, Princeton, NJ, USA. [2] Stevens Institute of Technology, Hoboken, NJ, USA. [3] Tufts University, Boston, MA, USA. [4] Potsdam Institute for Climate Impact Research, Potsdam, Germany. [5] Department of Earth & Planetary Sciences and Rutgers Institute of Earth, Ocean, and Atmospheric Sciences, Rutgers University, New Brunswick, NJ, USA. [6] US Army Corps of Engineers, Washington, DC, USA. [7] Vrije Universiteit Amsterdam, Amsterdam, The Netherlands. [8] Binera, Inc., Rockville, MD, USA. ✉email: bstrauss@climatecentral.org

For certain classes of extreme weather events, a growing body of research confers increasing confidence in probabilistic attribution of individual events to anthropogenic climate change[1]. Well-established classes so far include extreme temperatures, precipitation, and drought, while work on tropical and extratropical cyclones has begun but is not yet mature (e.g., refs. [2,3]). However, the flood impacts of all cyclonic coastal storms are amplified by a worsening factor that is quantitatively attributable to climate change: the anthropogenic component of sea level rise[4,5].

Linkage of an individual storm to climate change is a challenge[6], but logically unnecessary for attributing damages from sea level rise alone, which is a useful endeavor in itself. By a recent estimate, the global mean sea level (GMSL) increased $17.9 \pm 4.5$ cm ($1\sigma$) over 1900–2012 and this rise is continuing to accelerate[7]. Accordingly, all recent coastal floods start from a higher baseline water elevation than they otherwise would have, providing a direct physical basis for impact attribution.

Multiple studies document sea-level-rise-driven increases in the frequency (e.g., refs. [8,9]), peak height[10,11], and damages[12] of historical coastal floods. Most of this research assumes a simple linear addition of sea level rise to water levels, without any hydrodynamic modeling to capture observed nonlinear effects (e.g., refs. [13,14]). None of it, to our knowledge, isolates the effect of the climate-mediated human contribution to sea level rise from other factors such as natural variability and local vertical land motion. This human contribution can worsen damages, relative to what they would have been without it, regardless of how non-human factors influence total local relative sea level change prior to any chosen event.

Here, we investigate how much of the more than $60 billion in damages from Hurricane Sandy[15] can be linked to climate-related anthropogenic sea level rise (ASLR). We exclude from ASLR the effects of other components of sea level change, such as a net change in land-water storage (LWS) through groundwater extraction and surface water impoundment behind dams (which are anthropogenic but not caused by climate change), and such as local land subsidence (which may be anthropogenic but, when so, unrelated to climate change).

Sandy was a powerful hybrid (tropical/extratropical) cyclone, causing the highest water level in at least 300 years in the New York City metropolitan area[16]. This study neglects any other possible effects of climate warming aside from sea level rise (such as thermodynamic changes, potentially affecting storm track and intensity). Studies have so far found no evidence that Sandy's intensity, size, or unusual storm track were made more likely by climate change[17–19]. More broadly, a recent study found that future climate change effects on tropical cyclones will have only a small effect on extreme sea levels in New York Bight, relative to the effects of sea-level-rise[9,11,20]. Sandy had a worst-case timing with respect to the evening high tide[21] and a near-worst case storm track[22]. Any differences from this scenario in a hypothetical world without climate change would likely have reduced damages. Conversely, if climate change did influence Sandy's actual track and timing, it would likely have worsened damages. We thus seek to quantify a lower bound for the effect of anthropogenic climate change by focusing on the influence of linked sea level rise alone.

To do so, we first build on recent research to develop multi-method estimates for global and New York-area ASLR through 2012, when Sandy took place. We employ a high-resolution dynamic flood model and spatially varying error correction to simulate Sandy's peak flood both as it occurred, and under counterfactual conditions with lower baseline sea levels taken from throughout the range of ASLR estimates. We model damages based on these simulations, estimate the fraction that

can be attributed to ASLR by comparing results under actual vs. counterfactual conditions and estimate total attributable costs in the context of published Sandy impact estimates. Our study region is New York and the Atlantic coast of New Jersey—together, where most of the storm's damages took place—plus Connecticut. We find that across the full range of our estimates, climate-mediated ASLR drove multi-billions of dollars of damage from Hurricane Sandy. More broadly, this case study underscores that human-caused sea level rise has contributed to damages associated with other past coastal floods and will increasingly aggravate damages in the future as sea levels continue to rise, driven by anthropogenic warming. The approach demonstrated here may be applied to quantify those damages.

## Results

Results are presented as 50th (5th–95th) percentiles unless otherwise noted.

The link between anthropogenic climate change and global mean sea level rise is strongly established[23]. Here we employ largely independent approaches for attributing rise based either on reconstructed sea-level-rise budgets or on semi-empirical models of sea level as a function of global mean temperature. We develop multiple and integrated estimates for both total and attributable sea level rise both globally and for our study region, as represented by New York City, over the period 1900–2012. For New York, the quantity we estimate is sea level rise linked to contemporary climate change (climate-linked rise for short), omitting the effects of glacial isostatic adjustment through induced vertical land motion and changes in the geoid.

**Budget-based attribution of sea level rise**. Largely following ref. [24], we collate recent research literature to construct a global mean sea-level-rise budget for the study period, based on contributions from land-ice melt, ocean thermal expansion, and land-water-storage changes (Supplementary Table 3). We apply GRD sea level fingerprints (related to changes in Earth gravity, rotation, and deformation[25]) to each component and add a literature-based ocean-dynamics sea level term to build a New York budget. Both globally and for New York, the resulting totals are consistent with observations; see Supplementary Tables 1 and 2 and Fig. 1 panel a.

We then apply literature-based low, central, and high attributable fractions to each applicable global and New York budget component to develop a range of estimates for ASLR at each scale. Under the central scenario, which we take as approximating the most likely range of values, 9.8 cm (6.4–13.7 cm) of global mean sea level rise and 8.9 cm (5.2–13.1 cm) of New York-area rise are attributable to human causes mediated by climate change (Table 1).

Median ASLR for New York is 87%, 91%, and 92% of median global ASLR under low-, central- and high-attribution scenarios, respectively. The drivers for this consistent difference are fingerprint coefficients with values below one for mass losses from the Greenland ice sheet and global glaciers. Above-global-mean potential anthropogenic effects on New York sea level from Antarctic ice sheet mass loss and from ocean dynamics are not enough to offset below-global-mean contributions from Greenland and glaciers. The New York-to-global ASLR ratio forms an input for our next analysis.

**Semi-empirical attribution of sea level rise**. To develop semi-empirical estimates, we update ref. [5] (K16), which found that anthropogenic climate change drove at least 49% of 20th-century global mean sea level increase ($P \geq 0.95$). This fraction is consistent with[4,26,27] or lower than[28] previous findings.

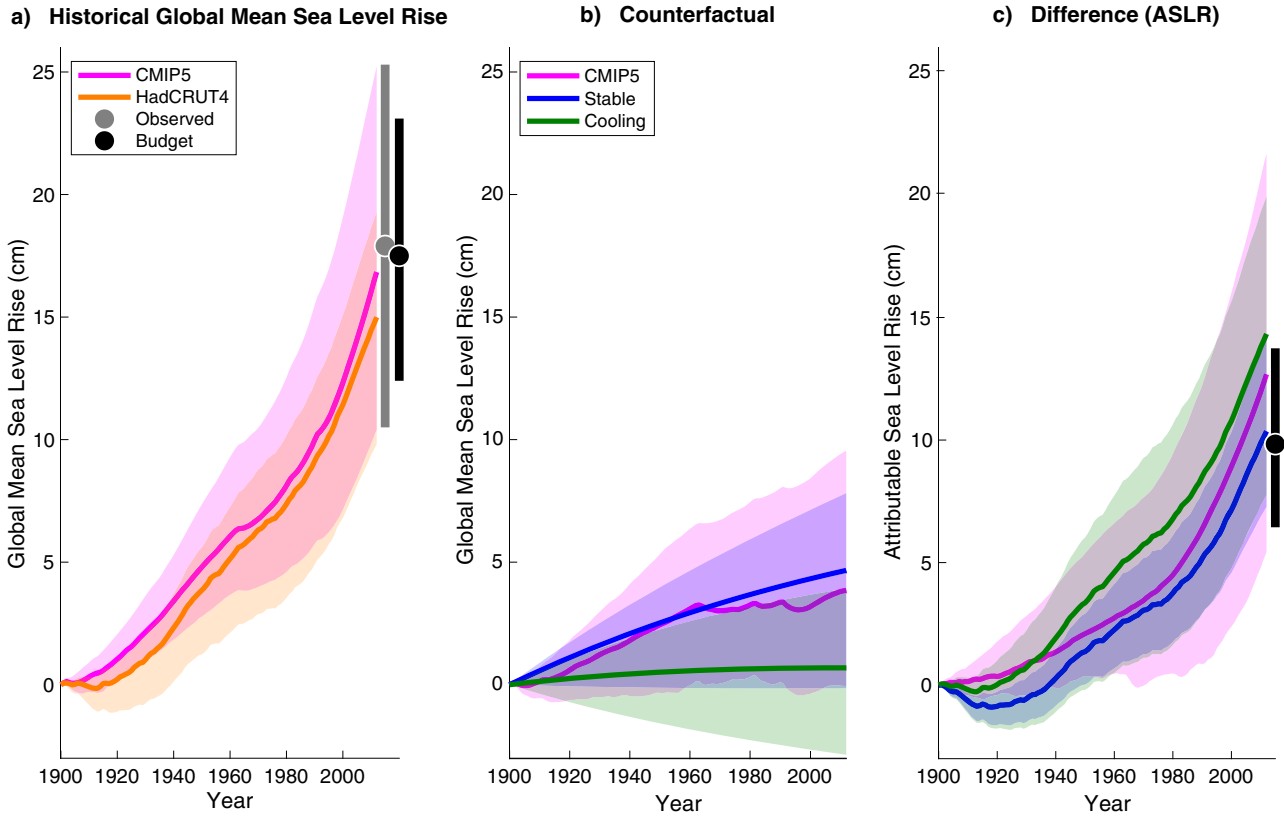

**Fig. 1 Modeled and attributable changes in global mean sea level (GMSL) from 1900 to 2012. a** Semi-empirically modeled historical GMSL estimates, based on global temperature timeseries from HadCRUT4 (observations) and from the CMIP5 historical scenario (simulations). **b** Semi-empirically modeled counterfactual GMSL estimates, from temperature-based stable or cooling counterfactual scenarios, and from the CMIP5 natural scenario. **c** Estimates of anthropogenic sea level rise (ASLR), based on differences between matching scenarios: HadCRUT4 vs. stable or cooling temperatures, and historical vs. natural CMIP5 simulations. Curves reflect medians of results pooled across semi-empirical model calibrations for both long-term temperature reconstructions[29,30]; shaded regions indicate 5th–95th percentile ranges. Also included are the medians (points) and 5th–95th percentiles (vertical bars) of historical (panel **a**) and central-case attributable (panel **c**) sea-level-rise budgets (Supplementary Table 3). Observed GMSL from ref. [7] over 1900–2012 (median and 5th–95th percentiles) is also shown for comparison (panel **a**; Supplementary Table 1).

**Table 1 Estimates of sea level rise attributable to anthropogenic climate change for 1900–2012 in cm.**

|  | Climate-mediated attributable sea level rise | |
| --- | --- | --- |
|  | **Global** | **New York** |
|  | **50th (5th–95th)** | **50th (5th–95th)** |
| *Budget reconstructions* | | |
| Low attribution scenario | 7.0 (4.1–10.4) | 6.1 (3.3–9.5) |
| Medium attribution scenario | 9.8 (6.4–13.7) | 8.9 (5.2–13.1) |
| High attribution scenario | 12.6 (8.7–17.1) | 11.6 (6.7–17.0) |
| *Semi-empirical modeling* | | |
| Stable temperature scenario | 10.4 (7.3–14.3) | 9.4 (6.6–13.0) |
| Cooling temperature scenario | 14.3 (7.8–20.0) | 13.0 (7.1–18.2) |
| CMIP5-based temperature | 12.7 (5.4–21.7) | 11.5 (4.9–19.7) |
| Semi-empirical ensemble | 12.0 (7.3–19.0) | 10.9 (6.6–17.3) |
| *Integrated summary* | | |
| Total ensemble | 10.5 (6.6–17.1) | 9.6 (5.6–15.6) |

K16 used a four-part methodology. First, based on their reconstruction of sea level over the last two millennia, they developed a semi-empirical model for global mean sea level with two alternative calibrations employing two alternative global mean surface temperature (GMST) reconstructions[29,30]. Second, using this semi-empirical model, K16 hindcasted sea level under

the observed HadCRUT4 global temperature record[31]. Third, they modeled global sea level under two different counterfactual twentieth-century global temperature scenarios—one in which 20th-century GMST was stable at the 500–1800 CE mean (stable scenario) and one in which it reverted to the 500–1800 CE cooling trend (cooling)—to quantify alternative natural histories without human influence. Finally, K16 compared the counterfactual and historical results over the 20th century to estimate anthropogenic contributions.

Here we apply the same methodology through 2012. Consistent with standard attribution methods[1], we also add an approach employing global mean temperature pathways built using historical and natural-forcing-only experiments from climate models participating in CMIP5. We pool simulations from stable, cooling, and CMIP5 approaches to develop semi-empirical ensemble results. To develop New York-area estimates, we multiply global-scale results by 91%, the New York/global relationship from our budget-based central scenario.

Modeled historical results are broadly consistent with sea level observations as well as with budget-based estimates (Supplementary Table 1 and Fig. 1 panel a), except that the HadCRUT4-based reconstruction of New York SLR is slightly low compared to observations (Supplementary Table 2). Semi-empirical estimates for both global and New York ASLR have a high overlap with budget-based findings (Table 1).

Because of the general consistency of results across major approaches (Fig. 1, Table 1) and in order to develop a single

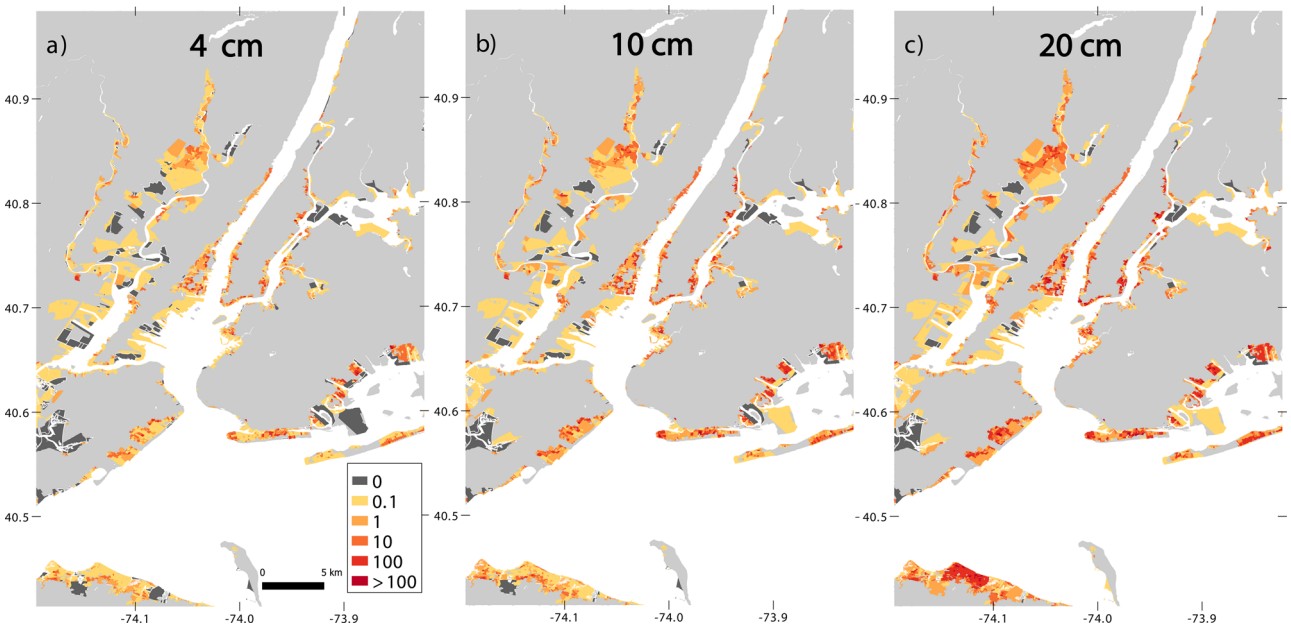

**Fig. 2 Differences in Census block-level modeled damages ($Million) between actual sea level and each of three counterfactual ones, for Manhattan and surrounding areas. a** 4 cm subceeds the 5th percentile value of all but one anthropogenic sea-level-rise estimate (see Table 1); **b** 10 cm roughly matches the total ensemble median; and **c** 20 cm exceeds the largest 95th percentile value. Legend values represent bin upper limits. For visibility, full blocks are shaded even when only a fraction of the block flooded.

overall characterization of ASLR that incorporates the most information possible, we pool semi-empirical and budget-based results with equal weights to develop total ensemble estimates for ASLR. The global total ensemble estimate is 10.5 cm (6.6–17.1 cm) and the New York-area estimate is 9.6 cm (5.6–15.6 cm) (Table 1).

**Damages**. New York, New Jersey, and Connecticut collectively reported more than $62.5 billion in repair, response, and restoration costs (with New York and New Jersey alone accounting for 96% of United States federal government disaster relief obligations nationwide[32]). The total repair and response damages claimed by New York, New Jersey, and Connecticut were $32.8 billion, $29.5 billion, and $360 million, respectively[15,33]. FEMA models indicate that wind caused less than 0.01% of damage in these states, so nearly all Sandy costs can be treated as coastal flood damages[34]. Moreover, riverine flood damages were likely negligible, as October 27–31 rainfall totals for most of the tri-state region were about 2.5 cm or lower (e.g., ref. [35]).

To develop damage estimates attributable to ASLR, we use hydrodynamic modeling and spatial bias correction to simulate Sandy's flood, first using actual sea level, setting up a baseline, and then as the flood might have unfolded given lower sea levels (see the "Methods" section). Hydrodynamic modeling of sea-level-rise impacts on storm-driven flooding is generally justified because many prior studies have shown that there can be nonlinear interactions between coastal floods and sea level rise (e.g., ref. [14]). In particular, a prior modeling study at New York City compared static and dynamic modeling methods and showed that non-linear effects are important for tropical cyclone flooding but less important for weaker storms such as winter nor'easters[13]. Spatial bias correction reduces the essentially unavoidable spatial correlation of error remaining in hydrodynamic model results.

Our baseline simulation achieves a close match to observed maximum flood levels (Supplementary Figs. 1 and 2). We next apply a common damage model to the simulated actual maximum flood-depth map and to simulated maps based on a

range of counterfactual lower starting sea levels (see e.g. Fig. 2, and the "Methods" section). Following ref. [15], we do not interpret modeled damages directly, but rather use them to estimate the relationships among damages in different locations and scenarios. We thus compute fractions of damages attributable to ASLR and apply these to total state-reported costs to develop dollar value estimates of attributable damages.

Based on total ensemble estimates for ASLR, 13% (7.5–23%) of Sandy damages in the tri-state area are attributable to climate-mediated anthropogenic sea level rise, amounting to $8.1B ($4.7B–$14B). Across estimates for ASLR from all six atomic approaches employed (three budget-based and three semi-empirical ones), $2.8B is the lowest estimate for attributable damages, corresponding to the 5th percentile of the low-attribution budget-based value. Table 2 breaks total ensemble results down by state, and Supplementary Fig. 3 presents the percentage of attributable damage at the county level. Supplementary Data 1 presents total, state, county, and New York City damages for all ASLR estimates. New York City, the sum of five New York counties, saw $1.5B ($0.9B–$2.5B) in damages attributable to total ensemble ASLR.

Our full method for damages estimation includes hydrodynamic simulation, spatial bias correction, and damage modeling. We also find that simpler approaches can serve as useful proxies (Supplementary Table 6 and Supplementary Fig. 4). A model with only hydrodynamic simulation and damage modeling (without bias correction), and a model with hydrodynamic simulation alone, using flood volume as a proxy for damage, have relative errors of −4.2% and 9.9%, respectively, compared to the full, ground-truthed model. These errors are calculated using the ratio of the percentage of the damage attributed to 50th-percentile total ensemble ASLR under each of the simplified models (hydro-dynamic simulation plus damage, or simulation alone) versus under the full model, as shown in Supplementary Table 6. Median estimates under each method fall within the 5th–95th percentile range of all other methods.

ASLR affects flood volume and property damage more strongly than simple spatial exposure of housing or population. This is

expected because increased exposure is a function only of the marginal increase in 'floodprint' affecting new areas, whereas greater flood depths are in effect across the full flooded area and thus influence volume and damage everywhere, as both are functions of depth (Fig. 3). We estimate that in the tri-state area, total ensemble ASLR exposed an additional 71 (40–131) thousand more people and 36 (21–66) thousand more housing units, 9.2% (5.3–17%) and 8.8% (5.0–16%, respectively (Table 2). Supplementary Data 2–4 give results for ASLR estimates for additional population, housing, and land exposure at the county, state, and tri-state levels, and for New York City.

**Table 2 Percentage of and total property damage (total in $Billions) and population (thousands) and housing exposure (thousands) attributable to anthropogenic sea level rise (ASLR) as estimated using total ensemble ASLR and full modeling with hydrodynamic simulation, spatial bias correction, and flood-depth-based assessment of property damage.**

|  | Attributable to ASLR | |
| --- | --- | --- |
|  | **Percent** 50th (5th–95th) | **Total** 50th (5th–95th) |
| *Tri-State Area* | | |
| Damage | 13.0% (7.5–22.5%) | $8.1 ($4.7–$14.0) |
| Population | 9.2% (5.3–17.1%) | 70.6 (40.4–131.0) |
| Housing | 8.8% (5.0–16.0%) | 36.3 (20.6–65.8) |
| *New York* | | |
| Damage | 13.3% (7.7–21.1%) | $4.2 ($2.4–$6.7) |
| Population | 9.5% (5.4–15.8%) | 45.0 (25.6–74.5) |
| Housing | 9.2% (5.2–15.3%) | 18.9 (10.7–31.2) |
| *New Jersey* | | |
| Damage | 12.8% (7.4–24.3%) | $3.7 ($2.2–$7.0) |
| Population | 8.8% (5.1–19.5%) | 24.5 (14.2–54.6) |
| Housing | 8.4% (4.8–16.8%) | 16.7 (9.5–33.5) |
| *Connecticut* | | |
| Damage | 10.4% (6.0–17.1%) | $0.18 ($0.10–$0.30) |
| Population | 6.8% (3.8–11.5%) | 1.1 (0.62–1.9) |
| Housing | 7.3% (4.0–12.4%) | 0.62 (0.35–1.1) |

## Discussion

Robust estimates for attributable sea level rise comprise a critical element of this analysis. We build confidence by employing independent methodologies, a best practice in current extreme event attribution science[36,37], by characterizing the distributions of all results, and by assessing their common ground. Confidence also comes from the consistent match between observations and budgeted/modeled total global mean sea level rise under the different approaches. Each median estimate falls within the 5th–95th percentile range for each other method and only the HadCRUT4-based median is more than 1.2 cm (7%) away from the observed one (Supplementary Table 1).

Estimates for total climate-linked sea level rise in New York largely overlap with observations as well, although not quite as robustly, with the central estimate of observed rise exceeding other median estimates by 2.4–4.8 cm (14–34%) (Supplementary Table 2). This excess may stem from inadvertent inclusion of some effects of land subsidence in the computation of the observed sea level rise total, despite the deduction made for glacial isostatic adjustment. The deduction may be biased low, or we may omit other causes of land subsidence at The Battery, the tide gauge location used for water level observations. Non-attributable short-term variability linked to ocean dynamics may also inflate observations (see below). If our budgeted and modeled estimates do in fact leave out a portion of climate-linked local sea-level rise, and if a fraction of this omission represents attributable rise, then this mismatch biases our overall estimates of ASLR and associated damages low.

Our global ASLR estimates are divided into two fully independent sets based on sea-level-budget reconstruction and semi-empirical modeling. Only the budget-based low-attribution estimate for ASLR appears statistically distinct from other ASLR estimates. Medians for the other five atomic approaches (two budget-based, three semi-empirical) all fall within each other's 5th–95th percentile intervals, with just one partial exception. (The range for the semi-empirical cooling scenario contains the median for the budget-based central-attribution scenario, but not vice versa.) The 90% interval from the total ensemble encompasses all median estimates and most 5th/95th percentiles, suggesting the total ensemble as constructed here is a reasonable and robust singular characterization of ASLR.

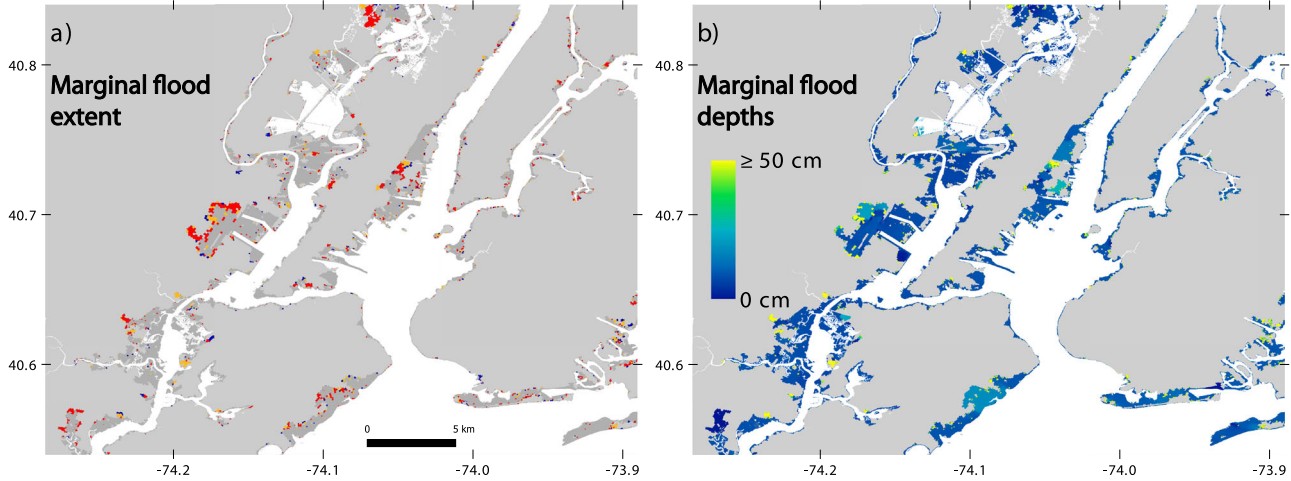

**Fig. 3 The marginal effects on flooding contributed by different amounts of anthropogenic sea level rise (ASLR). a** The estimated marginal contributions to flood extent for three sea level differences spanning 5th–95th percentile ASLR estimates from budget-based and semi-empirical approaches (Table 1). Hydrodynamic modeling indicates Sandy would not have flooded blue areas if the starting sea level had been 4 cm lower; neither blue nor orange areas if 10 cm lower; and neither blue, orange nor red areas if 20 cm lower. The slightly darkened gray areas would have flooded in any of these cases. **b** The estimated marginal contribution to flood depths for the sea level difference closest to the median total ensemble ASLR estimate (10 cm).

Applying the relationship found between global and New York ASLR in the budget approach to downscale our semi-empirical results reduces independence between approaches to a small degree. This choice assumes a similar underlying distribution of sea-level-rise components, with contributions from glaciers and the Greenland ice sheet substantially outweighing ones from the Antarctic ice sheet. This balance is well established for the 20th century and since[24]. The choice further assumes a similarly negligible contribution from ocean dynamics in differentiating New York-area sea level change from global change over the long term. Our confidence in this approach is strengthened by the consistency of the relationship between global and New York ASLR across low-, central-, and high-attribution scenarios in the budget-based approach.

The finding of lower ASLR in New York than globally might seem unlikely in light of prior studies presenting evidence for enhanced sea level rise along the U.S. mid-Atlantic coast driven by ocean dynamics over recent decades[38,39], perhaps via the slowing of the Atlantic Meridional Overturning Circulation[39–41]. However, more recent work indicates that mid-Atlantic sea level hot spots appear to be temporally cyclical and of short duration (below 10 years) and strongly related to indices of climate variability[42], consistent with the small net contribution of ocean dynamics to the New York sea level budget used here[43]. Short-term non-attributable dynamic increase in New York SLR may partially account for the slight mismatch between observed and modeled/budgeted climate-linked SLR noted previously and shown in Supplementary Table 2.

Other research and considerations suggest ASLR values toward the higher end of the distributions developed here. First, recent timeseries analyses employing purely statistical approaches suggest minimum (first percentile) values of 1.1 and 1.2 mm/yr for anthropogenic sea level rise over the century before Sandy for NE America[44] and for New York City[27], corresponding to at least 12.3–13.4 cm of ASLR for the focal period of the present study.

Second, in common with other attribution studies, these recent analyses and all but one of our own approaches assume a counterfactual natural 20th century with no temperature trend and no sea level trend. However, declining mean global temperatures from 500 CE until the onset of the industrial revolution in 1800 CE, consistent with the astronomically driven cooling trend that has characterized the 6000 years prior to the nineteenth century[45], suggest that cooling and thus lowering counterfactual may be more appropriate. A lower natural baseline would translate to higher values for ASLR, as in the cooling scenario (Table 1).

Third, including potential ASLR predating 1900[28], which we neglect here, would increase estimates slightly further; observations suggest ~0.4 cm or more per decade of total global mean sea level rise in the late 19th century[5,46].

Whatever the value of ASLR, total economic damages may also be greater than damages indicated here. Our estimates do not account for potential long-term economic effects, such as losses and gains in broad economic activity associated with employment and production changes across industries in the aftermath of a damaging cyclone strike[47]. Other studies of indirect damages find they may scale slightly exponentially with direct damages (e.g. exponent near 1.4[48]) but may stay relatively low even for exceptional events (e.g. roughly one-third of more than $100 billion in total damages for Hurricane Katrina[49]). Any nonlinear scaling would bias low our estimates of damages associated with ASLR, but this effect should be small, as our study concerns the difference between two very similar floods.

Sea level rise is robustly linked to climate change, providing a strong foundation for event and impact attribution. In this study, we advance progress in climate attribution by measuring the additional exposure and damage of Hurricane Sandy driven by sea level rise associated with anthropogenic climate change. We simulate coastal water levels, flooding, and damage in New York, New Jersey, and Connecticut, both as they occurred, and as they would have occurred across a range of lower sea levels corresponding to different estimates of attributable sea-level rise. The same general approach demonstrated here may be applied to other past and future coastal storms to estimate sea-level-linked anthropogenic climate damages from those events and improve estimates of the costs of climate change more broadly.

## Methods

### Sea-level-rise attribution

*Attribution from sea level budgets.* Drawing heavily on the framework and analyses of ref. [24], we build budgets of global mean and climate-linked New York sea level rise from literature-derived estimates of each contributing component during the study period (see Supplementary Table 3). Greenland ice-sheet (GrIS) mass loss, Antarctic ice-sheet (AIS) mass loss, glacier melt, thermal expansion, and net changes in land-water storage all contribute to sea level changes both globally and near New York (net LWS change is assumed not to be climate-linked but is included for comparisons to observations). Ice-sheet, glacier, and LWS contributions are localized to New York through their individual GRD sea level fingerprints, taken from ref. [50]. We treat steric sea level rise as a constant global mean, with any New York-area anomalies from this rise represented by a regional ocean-dynamics term adapted from ref. [43].

Attributable fractions for different components can span wide ranges of statistical confidence across the literature. To both overcome this challenge and illustrate this range, we develop three attribution scenarios: high, central, and low. These estimates are drawn from $+1\sigma$, median, and $-1\sigma$ literature-based estimates of attributable fraction, when applicable (e.g. mountain glaciers and thermal expansion). In cases where there are less confident attributable fractions (e.g. the AIS and ocean dynamics), our low and high estimates span a range of attribution from 0% to 100%. We assume that central estimates are more likely in these cases, as well as in the better-characterized ones, and use central values as our main and summary estimates for the budget approach.

Supplementary Table 3 details individual sea level components, their attributable fractions, their sea level fingerprints at New York, the literature sources and assumptions made for all estimates, and budget totals combining these elements into overall and attributable sea-level rise both globally and for New York.

*Attribution from semi-empirical models.* The semi-empirical sea level model of K16 relates the rate of global mean sea-level change $dh/dt$ to global mean temperature $T$ ($t$):

$$dh/dt = a(T(t) - T_0(t)) + c(t) \qquad (1)$$

where $a$ reflects sea level sensitivity to deviations from an equilibrium temperature $T_0(t)$, and $c(t)$ is a minor term representing the long-timescale response to much earlier, pre-anthropogenic climate changes. K16 describes this model, including function parameters, in more detail.

For both stable and cooling counterfactual scenarios (mean and linear scenarios in K16), we use 1000 parameter combinations for the semi-empirical model, representing the full posterior probability distribution in K16. To model the temperature uncertainty we combine each set of parameters with 100 temperature samples (drawn from an AR(1) distribution as described in detail in K16). Thus, we compute 100,000 sea level curves for each combination of counterfactual scenario and calibration scenario; or 200,000 curves for each scenario, pooling across calibrations. We develop corresponding historical curves based on the HadCRUT4 temperature timeseries, such that the differences (ASLR estimates) are computed only with counterfactual results using matched parameters and calibrations.

For the CMIP5-based simulations, all climate models with both historical and natural-forcing-only experiments were used (totaling 14 models and 43 unique realizations). We use only natural forcing for the counterfactual case and full forcing (including anthropogenic forcing) for historical cases[51]. For historical simulations, we use GMST from historical test runs where available and historical runs where not available. Because many of the historical test runs end in 2010 and historical in 2005, we extend these runs using emissions from representative concentration pathway (RCP) 8.5[52], which closely track actual emissions after these endpoints. For the counterfactual historicalNat experiment, which includes only natural forcing, 11 of the CMIP5 ensemble members end in 2005 and 32 in 2010; temperatures are linearly extrapolated to 2012 based on their trend over the last 50 years of data. For each CMIP5 ensemble member, we pair counterfactual and historical temperature timeseries. As we do with the purely temperature-based scenarios (stable, cooling, and HadCRUT4), we then use the parameter space from the K16 semi-empirical model to convert each paired temperature timeseries into 1000 paired sea level timeseries. We perform this procedure for both the calibrations of the semi-empirical sea-level model, from refs. [29] and [30]. The end result is 1000 paired sea level curves for each combination of CMIP5 realization (43) and semi-empirical calibration (2). Supplementary Table 5 provides sea level

and attribution results for all 86 combinations, as well as 43 summaries for each CMIP5 realization pooled across calibrations. Pooling across all realizations, $n = 43,000$ for each calibration, and pooling across all realizations and calibrations, $n = 86,000$, supporting results shown in Supplementary Table 4.

*Ensemble estimates*. We combine our budget-based and semi-empirical modeling approaches to develop a total ensemble. First, we randomly resample $n = 100$ million times (with replacement) from each of the three pooled semi-empirical global and New York ASLR scenario distributions (stable scenario, cooling scenario, CMIP5 scenario) to create a semi-empirical ensemble distribution, with results shown in Table 1. New York samples are scaled as 91% of the global semi-empirical samples, as described above. Next, to make use of the budget-based estimates, we assume that central-attribution-scenario estimates are most likely for all sea-level-rise components, including those with better or more poorly characterized attributable fractions. This choice is reinforced by the fact that summary statistics from the central scenario are very similar to those from pooling values from all three scenarios (high, central, and low). Global and New York central-estimate ASLR distributions are randomly sampled $n = 300$ million times (to match the number of pooled semi-empirical values, ensuring equal weighting across the two overarching approaches) from a skew-normal distribution constrained by the distribution 5th, 50th, and 95th percentiles. Finally, pooling across the semi-empirical and budget-based samples yields $n = 600$ million values each for global and New York ASLR; these comprise ensembles from which we draw our integrated summary statistics (Table 1).

**Flood modeling**. Hydrodynamic flood simulations are performed using the U.S. Army Corps of Engineers Coastal Storm Modeling System (CSTORM-MS)[53], exactly as performed for the North Atlantic Coastal Comprehensive Study[54] except using pre-2012 topography and bathymetry. CSTORM-MS includes the vertically integrated, two-dimensional (2D) hydrodynamic model ADCIRC (ADvanced CIRCulation model)[55–57], tightly coupled with the STWAVE wave model[58–61] for nearshore areas and coupled with the WAM wave model[62] for deepwater areas. Wind and atmospheric pressure forcing for Sandy are from a meteorological reanalysis created by OceanWeather, Inc., assimilating observations with atmospheric modeling to produce a highly accurate reproduction of the storm conditions. Streamflows for several rivers are included as input within the ADCIRC model (see ref. [54] for details).

The Sandy historical run is implemented with a mean sea level offset of +10 cm, as mean sea level for the year 2012 at The Battery, New York City, was +10 cm relative to the 1983–2001 tidal epoch. The ensemble of counterfactual runs includes mean sea level perturbations, relative to the historical run, of −24, −20, −16, −14, −12, −10, −8, and −4 cm. This captures a broad range of possible uncertainty in ASLR; we sample more densely near the center of the distribution.

**Model validation and error reduction**. We compare maximum water elevations from our baseline historical simulation ($h_{sim}^{0\,cm}$) to 456 maximum water level observations ($h_{obs}$) from high water marks, surge pressure gauge readings, and tide gauge readings (Supplementary Fig. 1) compiled by the USGS in study region[63,64]. We exclude high water marks rated by USGS as poor" quality or located in the wave-active FEMA V Zone, as we are modeling still water elevation. Data are compared to the modeled maximum elevations at the nearest wet grid point. Points that the model predicted to be dry were estimated by linear interpolation, or by a nearest neighbor outside the convex hull of observations.

The areas with flood damages during Sandy had $h_{obs}$ from 2.5 to 4.2 m, and our raw model results have a bias of −1.2 cm and root mean square error (RMSE) of 35.5 cm. However, there is a clear spatial correlation to the error (Supplementary Fig. 2, left panel). This is a common challenge with coastal flood modeling and can arise from multiple sources, including baroclinic processes not captured in a 2D model (e.g. a buoyant coastal current) or erroneous (or storm-impacted) bathymetry in the inlet to a lagoonal back-bay estuary, several of which exist along the region's coasts. A spatially coherent bias correction is utilized to correct for this component (see Supplementary Methods). The resulting bias-corrected modeled maximum water levels have a reduced mean bias of −0.1 cm and RMSE of 22.3 cm, and the error field has no visible spatial correlation remaining (Supplementary Fig. 2, right panel). We apply the same bias correction to our counterfactual simulations $h_{sim}^{(4\,cm - 24\,cm)}$. While correction does improve model accuracy, the effects on final results are small (see Supplementary Table 6).

**Damage analysis**. We create flood maps using modeled, bias-corrected water elevations and lidar-derived land elevation data with roughly 5 m horizontal resolution. Water depth fields are computed by subtracting land elevation values from water elevation fields. Land elevations come from lidar-based digital elevation models compiled and distributed by NOAA[65]. Land inundation surfaces are created by thresholding to positive values of depth and refined using connected components analysis to remove low-lying areas that are isolated from the ocean.

We assess the exposure of land, population, and housing, as well as water volume over land, within individual Census blocks using boundary, population and housing data from the 2010 U.S. Census[66], and sum these intermediate results to wider spatial scales (county, state, etc.) as necessary. In computing population and

housing unit exposure, we assume uniform density within each block, except zero density in wetland areas defined by the National Wetland Inventory[67]. We determine the dryland area of each block masked by the inundation surfaces, and multiply by dryland population/housing density within the respective blocks to compute total exposure. Total volume of water covering land is found by integrating water depth across dryland regions.

Property damage is calculated using the standard HAZUS-MH method from the U.S. Federal Emergency Management Agency[68], using the same approach as utilized in ref. [69], which is described in detail in their supplementary information. In short, the damage calculation is based on depth-damage curves, which denote the vulnerability of a specific asset to different flood depths in the form of a percentage of maximum flood damage (which is based on depreciated replacement costs). The curves are typically non-linear, rising sharply through low inundation depths and leveling off with higher ones, and are based on observational studies using insurance claims data from the National Flood Insurance Program[70]. This approach does not explicitly address variability in non-depth mechanisms of damage that may include floating debris, chemical contamination, erosion, behavioral responses, and water speed (except that it does use adjusted curves in strong wave action zones). However, meticulous absolute damage estimates are not essential to our analysis; rather, we aim to estimate relative damages among simulated floods starting from slightly different baseline sea levels, and this goal aligns well with the emphasis on depth in the HAZUS-MH model.

Both depth-damage curves and maximum damages are differentiated for 33 building types ranging from various residential types (single home, multi-story apartment buildings, etc.) to, amongst others, industrial, commercial, and educational buildings. Counts of these building types are available per census block from the HAZUS databases. Therefore, we compute the mean inundation depth for flooded areas within each Census block, use this depth to compute hypothetical full-block damage for all buildings present in the Census block, and reduce this total by the fraction of the area not flooded. We assume that all buildings (and types) are distributed equally within the Census block.

Damages and exposures for precise amounts of ASLR (i.e., different percentiles of different budget-based, semi-empirical, or integrated scenarios) are interpolated between results for various water levels (from 4, 8, 10, 12, 14, 16, 20, and 24 cm of ASLR). Figure 2 shows block-level differences in modeled damages for three of these increments, each vs. actual sea level.

In addition to our full approach (hydrodynamic simulation, spatial bias correction, and damage modeling), we also test simpler methods which might be useful when detailed modeling is not feasible, including assessment of flood volume per block as a proxy for damage.

Following ref. [15], we do not interpret modeled damages directly, but rather use them to estimate the fraction of damages attributable to ASLR. We then apply the fraction against total state-reported costs to develop dollar value estimates of attributable damages. We distribute state totals among counties in proportion to counties' modeled damages.

**Reporting summary**. Further information on research design is available in the Nature Research Reporting Summary linked to this article.

## Data availability
Source data files (underlying each Table and Figure), semi-empirical model simulation results, hydrodynamic simulation results, and block-level depth and damage estimates are publicly available and archived at https://doi.org/10.5281/zenodo.4543662.

Publicly available datasets used as direct inputs are available as follows: HadCRUT4 annual mean global temperature reconstructions from www.metoffice.gov.uk/hadobs/hadcrut4/data/current/download.html; CMIP5 global temperature simulations from esgf-node.llnl.gov/projects/CMIP5/; parameter distributions for the semi-empirical sea level model, dataset S01(j) from pnas.org/content/suppl/2016/02/17/1517056113.DCSupplemental; global mean sea level timeseries reconstruction from ref. [7] Supplementary data 1, static-content.springer.com/esm/art%3A10.1038%2Fs41558-019-0531-8/MediaObjects/41558_2019_531_MOESM2_ESM.txt; New York City sea level observations from tidesandcurrents.noaa.gov/sltrends/sltrends_station.shtml?id=8518750; tide gauge uncertainty estimates from NOAA Technical Report NOS CO-OPS 053 53, Fig. 30, tidesandcurrents.noaa.gov/publications/Tech_rpt_53.pdf; glacial isostatic adjustment estimates from spatio-temporal modeling of ref. [71] from Supporting information Table S1, agupubs.onlinelibrary.wiley.com/action/downloadSupplement?doi=10.1002%2Fgrl.50781&file=Kopp2013_SI.pdf; and USGS data on Sandy high water levels from https://doi.org/10.3133/sir20155036 and https://doi.org/10.3133/sir20165085.

## Code availability
Hydrodynamic simulation was all performed with publicly available software: ADCIRC version 52.30, STWAVE version 6.2.28, and CSTORM Coupling Software version 1.1.16. Custom Matlab (R2017b+R2018b) code for performing spatial bias corrections, semi-empirical analyses, and observational analyses are documented and publicly available for download at https://github.com/climatecentral/cc_Sandy_matlab. Methods related to assessing land, population, and housing exposure were implemented using custom Matlab (R2017b), Python, and C++ code. Due to licensing restrictions by Climate Central, this code is not publicly available. However, these tasks can be accomplished

with standard GIS software. Damage assessment was performed using HAZUS-MH software version MR4.

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

## Acknowledgements

Benjamin Strauss, Scott Kulp, and Maya Buchanan were supported by the Kresge Foundation and the George and Estelle Sands Foundation. Benjamin Strauss, Robert Kopp, Scott Kulp, and Daniel Gilford were supported by NSF grant ICER-1663807 and NASA grant 80NSSC17K0698. Philip Orton was supported by NOAA award NA15OAR4310147 and NSF award 1855037. We thank Andrew Cox at Oceanweather Inc. for providing their meteorological reanalysis data for Hurricane Sandy, Chris Schubert, and Tom Suro (USGS) for providing data and interpretive guidance concerning maximum flood elevations. We thank Michael Oppenheimer for thoughtful comments on the manuscript. We acknowledge the World Climate Research Programme's Working Group on Coupled Modeling, which is responsible for CMIP, and we thank the climate modeling groups (listed in Supplementary Table 5) for producing and making available their model output. For CMIP, the U.S. DOE's Program for Climate Model Diagnosis and Intercomparison provides coordinating support and led the development of software infrastructure in partnership with the Global Organization for Earth System Science Portals.

## Author contributions

B.H.S. conceived the project. B.H.S., P.M.O., and R.E.K. oversaw analysis. K.B., D.M.G., C.M., S.V., S.K., and H.d.M. performed analysis. S.K, M.K.B., and D.M.G. prepared figures and tables. B.H.S., P.M.O., R.E.K., S.K., H.d.M., M.K.B., and D.M.G. wrote the paper.

## Competing interests

The authors declare no competing interests.
