## [Peer Review File · Nature Communications]

Reviewers' comments:

Reviewer #1 (Remarks to the Author):

Review of "Economic Damages from Hurricane Sandy Attributable to Sea Level Rise Caused by Anthropogenic Climate Change".

This paper seeks to attribute damages from Hurricane Sandy specifically to anthropogenic sea level rise. From a high level point of view, I fully support the objective of this paper. Putting a cost to ongoing sea level rise is an important "next step", and allows us to move on from talking about the cost of sea level rise in abstraction. This study is indeed novel in how it ties different pieces together and will likely generate broad interest. I do have some specific comments that I feel should be addressed before the paper is suitable for publication, although I am supportive of this study in general.

1. Section 2: There is an extended discussion on how the sea level rise in the Mid-Atlantic was attributed to anthropogenic forcing. This relies very heavily on prior studies, which is certainly not a problem. In the second to last paragraph, there is a brief discussion regarding natural variability and how it may have contributed to higher sea level along the mid-Atlantic. I agree that there is little conclusive, but it does seem like such signals could potentially impact the results in the paper and should be discussed in more detail. Indeed, the mid-Atlantic has been heavily studied in terms of sea level, with several papers focusing on the "hotspot" of sea level rise in this area. To essentially dismiss these as inconclusive, while stating that a global estimate is sufficient for New York is problematic and should be dealt with a clearer way.

2. How has land subsidence been accounted for in this analysis? Specifically, when attributing some portion of the past rate to anthropogenic sea level rise, is it necessary to remove the signal associated with vertical land motion? If an area was experiencing uplift, for example, presumably the impact of anthropogenic rise as computed here would be lessened? Perhaps just some further explanation would suffice here.

3. "Pre-warp" vs. "Post-warp": I am unfamiliar with the use of "warp" in this context. I generally assume this is my own ignorance, but I'm guess others may feel the same way - I suggest either defining the term or using some different. Regarding Fig. 3, what is the reason for the remaining large residuals in some locations? In many areas the residuals appear to be zero, but > 1 m in others. I understand the technique being employed here to reduce the residuals, but some explanation about why large residuals remain would be helpful.

4. The terminology in the paper is unlikely to be accessible to the broader population not familiar with this field. I defer to the editor if changes might be needed, but generally I assume Nature Communications would like to appeal to a broad audience. Perhaps more clearly stating what is meant in some cases (as in "warp" and "counterfactual case"). Not a huge issue, just a recommendation.

Reviewer #2 (Remarks to the Author):

The paper "Economic Damages from Hurricane Sandy Attributable to Sea Level Rise Caused by Anthropogenic Climate Change" is, from my Point of view, not appropriate for publication in Nature Communications. Specifically, the authors use a semi-empirical model to separate anthropogenic and natural forcing from observed global mean sea level rise and assume that the fraction also holds locally for the waters in front of New York. They justify this assumption by (i) saying that the CMIP5 modelled fraction of global/local sea-level in Messignac et al. (2018) would also be 1/1, (ii) that future

projections (dominated by anthropogenic forcing) would also project a 1/1 fraction (Kopp et al., 2014), and (iii) stating that there is too limited evidence for anthropogenically forced AMOC-related sea level change along this coastline over the 20th century from observations. I find this justification really weak:

1. First of all, a 1/1 ratio in the total rate of sea level change does not necessarily mean a 1/1 ratio for the anthropogenic contribution as the local budget of individual contributions (i.e. dynamic sea level, ice-melt, etc.) can and will significantly differ from the global one. Hence, the local anthropogenic contribution critically depends on the involved processes and can significantly differ from the global even if the total sea level change is not statistically different.

2. Second, the literature compilation presented here as a foundation for their statement is highly biased and neglects several relevant papers discussing the anthropogenic contribution to different sea-level components at local and global scale (much better summarized in the review article of Marcos et al, 2017: https://link.springer.com/chapter/10.1007/978-3-319-56490-6_15): For instance, for the dynamic component Marcos and Amores (2014, <https://agupubs.onlinelibrary.wiley.com/doi/full/10.1002/2014GL059766>) find an anthropogenic fingerprint of about ~ 87% since 1970 in global sea-level rise, while in the Atlantic anthropogenic forcing only accounts for about 65%. Consequently, the ratio for this component is already only 0.75 which is significantly different to the 1/1 relationship considered by the authors.

3. Third, I can't follow the argument that the authors' assumption is justified by a historically modelled CMIP5 ratio in total sea-level of 1/1 (Meyssignac et al., 2018) : If this ratio is correct it differs significantly from that in the real world, which suggests that New York sea levels have been rising much faster than the simultaneous GMSL (Sallenger et al., 2012). In consequence, this points to a significant mismatch between observations and models.

4. Fourth, the argumentation that there is no compelling evidence for a significant anthropogenic AMOC fingerprint in the observations does not justify that there is no other anthropogenic signature in other local dynamic components influencing the sea level at New York (it is still open how much the AMOC indeed explains of the dynamics along the US east coast).

Summarizing, the estimation of the anthropogenic contribution at New York is neither convincing nor justified (and also not state-of the art). The authors could do it much better by going the next step and applying a proper Detection & Attribution algorithm for each sea-level component at a local scale (the models have all been published).

More specific comments are as follows:

Abstract

Define HAZUS fist or write it in a different way, otherwise the reader has no idea what that is...

"We find that anthropogenic sea level rise (ASLR) is most likely responsible for approximately \$11.1B..."

Here information on the total damages (or as proportion of it caused by ASLR) attributable to Sandy would be useful.

1 Introduction

"Well-established classes so far include extreme temperatures, precipitation and drought, but neither

tropical nor extratropical cyclones..."

I am not entirely sure if this is correct. There was a paper by Dangendorf et al. (in ERL something between 2014 and 2016) which investigated whether extreme (extratropical) North Sea storm Xaver and the resulting surge was caused (or at least amplified) by climate change effects... Please check and consider if this might be a useful ref. Otherwise your reporting seems to be incomplete.

"However, the flood impacts of cyclonic coastal storms are amplified by one worsening factor strongly and quantitatively linked to climate change: global sea level rise"

Yes and no: SLR is certainly the main driver of increasing coastal extremes, but additional direct (changes in the meteor. forcing; stratification, etc.) as well as "indirect" (changes in tides) effect can amplify flood impacts.

"In this case, linkage of an individual storm to climate change is logically unnecessary for attributing damages..."

The first part of the sentence may refer to my last comment and would then be easier to follow...

"Only limited locations with sparse populations meet this description, along parts of Greenland, Antarctica, and Alaska, mainly due to reduced local gravitational pull on the ocean because of diminished local land-ice mass..."

Mainly due to glacial isostatic adjustments (GIA). Check that!!! See e.g. Peltier

"Most of this research assumes a simple 1:1 addition of sea level rise..."

Maybe „linear“ would be a better description

"...without any hydrodynamic modeling to capture known potential nonlinear effects (e.g., Orton et al, 2015; Arns et al, 2017)."

Well... not always known and still not 100% described by physics. I would delete the term „known“ or replace it by „observed“

"None of it isolates the effect of the climate mediated human contribution to sea level rise from other factors such as natural variability and local land subsidence."

At least none of the above mentioned... But there was a study (amongst others but i only remember this one) by Dangendorf et al. 2014 called evidence for long term memory in sea level discussing the anthropogenic effects on SLR.

"This study neglects any other possible effects of climate warming aside from sea-level rise (such as thermodynamic changes, potentially affecting storm track and intensity)."

...as well as changes in hydrodynamic properties (see e.g. Schindelegger et al., 2018 on changing M2 tides).

"Hurricane Sandy had a worst-case timing with respect to the evening high tide (Georgas et al, 2014) and a near-worst case storm track (Hall and Sobel, 2013), so any climate-related changes in the track and timing would likely have worsened damages."

No! I don't understand your statement (without describing it in more detail). Damages could also be less intense if the timing or tracks have changed.

3 Flood modelling

"The ensemble of counterfactual runs includes mean sea level perturbations of -24, -20, -16, -14, -12, -10, -8, and -4 cm, relative to the historical run."

Where do these numbers come from? Maybe I did not get it from above but it definitely needs to be explained!

4 Model validation and error reduction

"We exclude high water marks rated as \poor" quality or located in the wave-active FEMA V Zone..."

Where rated? By whom? Which criteria?

5 Damage analysis

"Property damage is calculated using the using the..."

Delete the second "using the"

"Damages and exposures for precise amounts of ASLR (e.g., the 50th, 5th, and 95th percentiles of the pooled CMIP5-, Stable-, and Cooling-based scenarios) were interpolated between results for various water levels (from 4, 8, 10, 12, 14, 16, 20, and 24 cm of ASLR)."

Until now I have not really understood what your ASLR runs are based on. You state above that it is the counterfraction runs assuming cooling and stable conditions. Where do 4-24 cm of ASLR come from? Maybe reword some above sections and try making it easier to follow for those not involved in the work.

6 Results and discussion

"These errors were calculated using the ratio of percentage of damage attributed to ASLR under the S+D or S models in SI Table 2, respectively, relative to under the S+BC+D model (all based on the 50th percentile from CMIP5)."

No chance understanding this... what is D? What is BC?

Reviewer #3 (Remarks to the Author):

The article sets out to quantify the contribution of anthropogenic sea level rise to the impacts of Sandy. The paper is overall well written and interesting, but at the moment I find that it has some major drawbacks which could even justify rejection for a high impact journal like NCOMMS. I explain better below:

The authors make clear that they focus on ASLR and not climate change in general. They consider previously published SLR projections, but one weakness of the study is that there is no mention on how climate change affected the probabilities to have an event like Sandy occur. I can understand that addressing this point may not be easy but in my opinion it is at least essential to discuss.

If I understand correctly the authors use a model which estimates global mean SLR. This implies the that MSL changes in New York follow the global mean. If this is the case is there any evidence to support this assumption?

I believe there is a lot of space for improvement in the description of the methods and data used. I include a lot of objections for the flood modeling in the paragraphs below. Also the SLR projections have been published before but the current information is not sufficient to follow the work done. The reader is forced to read the PNAS paper to be able to obtain essential information needed to understand the current study.

But the weakest part of the work in my opinion is in the flood modeling. First of all the authors apply bias correction to the model (eq 1) which I find alienating. Error in a hydrodynamic model is normally a reason to do better calibration and is not something that can be solved with bias correction. The correction adds bias in the results of a process based model which is a problem when different scenarios need to be explored (as is presently the case). Given that the area has very good coverage of tide and wave gauges these could allow for a detailed validation/calibration of waves, water levels along the domain. Having hydrodynamic processes properly simulated by the model would be a precondition for accurate flood modelling. Then model errors could be dealt with calibration, more accurate surface roughness data, etc.

All that background information is missing from the paper I think that the authors should show clearly that they did their best to oppress model errors before opting for bias correction. Similarly I don't understand why they chose to exclude tidal gauges inside wave action instead of pre-processing the time series to remove the wave effects (e.g. some low pass filter?).

It is mentioned that the modelled water levels are combined with LIDAR data. Does this mean that the LIDAR data are not included in the inundation simulations? Where they simulations or the authors just applied a bathtub method. If there was simulation the resolution in land was 70 m as stated? If yes along the whole subaerial part of the study area? All these need to be better clarified in the methods. Also I am surprised that there is no validation of the model in terms of flood extents. As far as I know flood maps from Sandy are available.

Regarding the damage analysis, again there are a lot of vague points. What is the resolution finally considered? Inundation depths are averaged for each census block, that should decrease the resolution and affect the accuracy of the estimates. One would expect a more detailed approach since a single event is studied and the spatial extent is rather limited. Another important limitation is that there is no consideration for indirect impacts which could imply a 50% or more underestimation of losses and distorts the results due to the fact that the relationship between direct and indirect losses is not linear. All the above are serious shortcomings and they are not even discussed while the methodology is very poorly described.

Another issue relates to the figures which I don't find informative, especially given the fact that the article has been submitted to a high impact factor journal. Figures 2 and 3 could improve aesthetically but most important seem to be more methodological and thus more appropriate for the SI. There are no spatial maps comparing flood extents, and most importantly the distribution of damages and how these vary among the compared scenarios. Apart from the tables and Fig 1 there is no display of uncertainty and confidence levels. It seems to me that the authors did not put the amount of work needed to the manuscript.

Finally, I found the literature review outdated. The authors cite one paper from 2017 and one from 2018, while during these past years there have been several interesting contributions related to coastal hazards and impacts including many in high impact journals (addressing future changes in climate extremes, developments in coastal flood modelling and their implications for coastal losses).

Reviewer #1 (Remarks to the Author):

Review of "Economic Damages from Hurricane Sandy Attributable to Sea Level Rise Caused by Anthropogenic Climate Change".

This paper seeks to attribute damages from Hurricane Sandy specifically to anthropogenic sea level rise. From a high level point of view, I fully support the objective of this paper. Putting a cost to ongoing sea level rise is an important "next step", and allows us to move on from talking about the cost of sea level rise in abstraction. This study is indeed novel in how it ties different pieces together and will likely generate broad interest. I do have some specific comments that I feel should be addressed before the paper is suitable for publication, although I am supportive of this study in general.

1. Section 2: There is an extended discussion on how the sea level rise in the Mid-Atlantic was attributed to anthropogenic forcing. This relies very heavily on prior studies, which is certainly not a problem. In the second to last paragraph, there is a brief discussion regarding natural variability and how it may have contributed to higher sea level along the mid-Atlantic. I agree that there is little conclusive, but it does seem like such signals could potentially impact the results in the paper and should be discussed in more detail. Indeed, the mid-Atlantic has been heavily studied in terms of sea level, with several papers focusing on the "hotspot" of sea level rise in this area. To essentially dismiss these as inconclusive, while stating that a global estimate is sufficient for New York is problematic and should be dealt with a clearer way.

We have extensively revised our approach to develop estimates of attributable sea level rise specific to New York, addressing the main concern here. Those estimates are consistent with little to no attributable contribution from ocean dynamics in the mid-Atlantic. We still briefly raise the issue of potentially above-average mid-Atlantic sea level rise in paragraph 5 of our Discussion (lines 220-229), and add a new reference (Frederikse et al 2017) buttressing our case that it is largely or wholly not attributable, and supporting our quantitative treatment of ocean dynamics (see Supplementary Table 3 and its notes).

2. How has land subsidence been accounted for in this analysis? Specifically, when attributing some portion of the past rate to anthropogenic sea level rise, is it necessary to remove the signal associated with vertical land motion? If an area was experiencing uplift, for example, presumably the impact of anthropogenic rise as computed here would be lessened? Perhaps just some further explanation would suffice here.

All approaches used for estimating sea level rise, both globally and for New York, and for both total rise and its attributable fraction, exclude vertical land motion (VLM). (The budget-based approaches do not include any VLM component; and the semi-empirical approaches are based on the relationship between global mean temperature and sea level.) Our source for observed global mean sea level rise, used only for comparison to results, factored out VLM effects on the tide gauge record. And we remove the modeled effect of glacial isostatic adjustment from the

New York tide gauge record at The Battery, again used only for comparison (our approach is described in Supplementary Table 2). Glacial isostatic adjustment is the principle cause of VLM at the Battery.

Lines 85-87 of the manuscript explain, "For New York, the quantity we estimate is sea level rise linked to contemporary climate change ('climate-linked' rise for short), omitting the effects of glacial isostatic adjustment through induced vertical land motion and changes in the geoid."

3. "Pre-warp" vs. "Post-warp": I am unfamiliar with the use of "warp" in this context. I generally assume this is my own ignorance, but I'm guess others may feel the same way - I suggest either defining the term or using some different.

We have eliminated "warp" and use "spatial bias correction" (or shortened to "bias correction" or "correction") instead.

Regarding Fig. 3, what is the reason for the remaining large residuals in some locations? In many areas the residuals appear to be zero, but > 1 m in others. I understand the technique being employed here to reduce the residuals, but some explanation about why large residuals remain would be helpful.

Figure 3 in our original submission is now Supplementary Figure 2. We add a brief discussion in Supplementary Methods about large residuals in the final paragraph of the Flood Modeling section (lines 85-92): "For a few locations, large model vs. observation differences remain even after the correction process (designed not to over-t the spatial error eld). A large difference can arise if there is a wall or tide gate blocking water flow that is not resolved in the model. Alternatively, observational error may be at play. For example, in some locations outside of V-zones (where wave action is more likely to play a role), some high water mark data differ from nearby USGS high water marks by 1 meter even when the data are not marked low quality. Observation error seems likely in these cases. This could arise if there were a debris line that wasn't a high water mark, for example, due to water runoff."

4. The terminology in the paper is unlikely to be accessible to the broader population not familiar with this field. I defer to the editor if changes might be needed, but generally I assume Nature Communications would like to appeal to a broad audience. Perhaps more clearly stating what is meant in some cases (as in "warp" and "counterfactual case"). Not a huge issue, just a recommendation.

We have eliminated "warp" as a term. We feel the meaning of "counterfactual" is fairly self-evident and it is also a term used in the literature on attribution of extreme weather events to climate change, to which this paper closely relates. However, responding to the reviewer's concern, we provide strong context to the term, underscoring its meaning, in each of its first three appearances (lines 67, 70, and 115).

Reviewer #2 (Remarks to the Author):

The paper "Economic Damages from Hurricane Sandy Attributable to Sea Level Rise Caused by Anthropogenic Climate Change" is, from my Point of view, not appropriate for publication in Nature Communications. Specifically, the authors use a semi-empirical model to separate anthropogenic and natural forcing from observed global mean sea level rise and assume that the fraction also holds locally for the waters in front of New York. They justify this assumption by (i) saying that the CMIP5 modelled fraction of global/local sea-level in Messignac et al. (2018) would also be 1/1, (ii) that future projections (dominated by anthropogenic forcing) would also project a 1/1 fraction (Kopp et al., 2014), and (iii) stating that there is too limited evidence for anthropogenically forced AMOC-related sea level change along this coastline over the 20th century from observations. I find this justification really weak:

We have greatly changed and expanded our approach to estimating attributable sea level rise in the New York area in response to this reviewer's concerns and are grateful, as we believe the changes have resulted in a substantially stronger piece of research. Specifically, we have developed a budget-based approach to estimating ASLR for New York, and we have leveraged the same to downscale our semi-empirical estimate as well. This change was the main driver for a major overhaul of the manuscript.

1. First of all, a 1/1 ratio in the total rate of sea level change does not necessarily mean a 1/1 ratio for the anthropogenic contribution as the local budget of individual contributions (i.e. dynamic sea level, ice-melt, etc.) can and will significantly differ from the global one. Hence, the local anthropogenic contribution critically depends on the involved processes and can significantly differ from the global even if the total sea level change is not statistically different.

This is a fair point. No such assumption is made in our new methodologies.

2. Second, the literature compilation presented here as a foundation for their statement is highly biased and neglects several relevant papers discussing the anthropogenic contribution to different sea-level components at local and global scale (much better summarized in the review article of Marcos et al, 2017: https://link.springer.com/chapter/10.1007/978-3-319-56490-6_15): For instance, for the dynamic component Marcos and Amores (2014, <https://agupubs.onlinelibrary.wiley.com/doi/full/10.1002/2014GL059766>) find an anthropogenic fingerprint of about ~ 87% since 1970 in global sea-level rise, while in the Atlantic anthropogenic forcing only accounts for about 65%. Consequently, the ratio for this component is already only 0.75 which is significantly different to the 1/1 relationship considered by the authors.

Thank you for pointing out these papers. Both are now integral parts of our budget-based attribution estimates, as detailed in Supplementary Table 3 and its notes.

3. Third, I can't follow the argument that the authors' assumption is justified by a historically modelled CMIP5 ratio in total sea-level of 1/1 (Meyssignac et al., 2018) : If this ratio is correct it differs significantly from that in the real world, which suggests that New York sea levels have been rising much faster than the simultaneous GMSL (Sallenger et al., 2012). In consequence, this points to a significant mismatch between observations and models.

We no longer assume that attributable SLR in NY is the same as attributable global SLR, and so no longer cite Meyssignac to justify such an assumption. Separately, we address the issue of potentially above-average mid-Atlantic sea level rise in paragraph 5 of our Discussion (lines 220-229), and add a new reference (Frederikse et al 2017) buttressing our case that it is largely or wholly not attributable, and supporting our quantitative treatment of ocean dynamics (see Supplementary Table 3 and its notes).

4. Fourth, the argumentation that there is no compelling evidence for a significant anthropogenic AMOC fingerprint in the observations does not justify that there is no other anthropogenic signature in other local dynamic components influencing the sea level at New York (it is still open how much the AMOC indeed explains of the dynamics along the US east coast).

This is a fair point. Our new approach addresses local ocean dynamics holistically and relies on Frederikse et al (see reply to your previous point).

Summarizing, the estimation of the anthropogenic contribution at New York is neither convincing nor justified (and also not state-of the art). The authors could do it much better by going the next step and applying a proper Detection & Attribution algorithm for each sea-level component at a local scale (the models have all been published).

We believe that developing a full Detection & Algorithm model for New York sea level is beyond the scope of this paper. Rather, we have found that existing literature is sufficient for estimating a global sea level rise budget over the study period that is consistent with observations (following the approach of the IPCC's Special Report on the Ocean and Cryosphere in a Changing Climate); for applying GRD sea level fingerprints to each component, so as to develop New York-specific estimates; for estimating attributable fractions of each component, particularly the biggest two components, glacier melt and ocean thermal expansion; and for developing estimates of New York-specific ocean dynamic contributions.

We believe and hope you will agree that pursuing these steps has significantly strengthened this manuscript.

We also note that because we have modeled the impact of a wide range of different anthropogenic SLR amounts, if new estimates of ASLR for New York are published in the future, based on evolving methodologies, it will be straightforward to update this study's results

accordingly, interpolating from the raw values we have developed for 4-24 cm and included in a public data depository.

More specific comments are as follows:

Abstract

Define HAZUS first or write it in a different way, otherwise the reader has no idea what that is...

We have removed "HAZUS" from the Abstract. Later, when HAZUS is first introduced in the text, we clarify, "Property damage is calculated using the standard HAZUS-MH method from the U.S. Federal Emergency Management Agency..." (lines 382-383)

"We find that anthropogenic sea level rise (ASLR) is most likely responsible for approximately \$11.1B..."

Here information on the total damages (or as proportion of it caused by ASLR) attributable to Sandy would be useful.

The first sentence of the abstract provides context of total damages.

1 Introduction

"Well-established classes so far include extreme temperatures, precipitation and drought, but neither tropical nor extratropical cyclones..."

I am not entirely sure if this is correct. There was a paper by Dangendorf et al. (in ERL something between 2014 and 2016) which investigated whether extreme (extratropical) North Sea storm Xaver and the resulting surge was caused (or at least amplified) by climate change effects... Please check and consider if this might be a useful ref. Otherwise your reporting seems to be incomplete.

Text revised to read, "Well-established classes so far include extreme temperatures, precipitation and drought, while work on tropical and extratropical cyclones has begun but is not yet mature..." with appropriate citations (lines 28-29).

"However, the flood impacts of cyclonic coastal storms are amplified by one worsening factor strongly and quantitatively linked to climate change: global sea level rise"

Yes and no: SLR is certainly the main driver of increasing coastal extremes, but additional direct (changes in the meteor. forcing; stratification, etc.) as well as "indirect" (changes in tides) effect can amplify flood impacts.

"In this case, linkage of an individual storm to climate change is logically unnecessary for attributing damages..."

The first part of the sentence may refer to my last comment and would then be easier to follow...

This text has been revised to make our point more simply and clearly: "However, the flood impacts of all cyclonic coastal storms are amplified by a worsening factor that is quantitatively attributable to climate change: the anthropogenic component of sea level rise." (lines 30-32)

“Linkage of an individual storm to climate change is a challenge, but logically unnecessary for attributing damages from sea level rise alone, which is a useful endeavor in itself.” (next paragraph, lines 33-34)

We also note here that we do capture the effect of sea level rise on tides, for the most part, with our hydrodynamic modeling. In response to another review comment, we have added a paragraph to the Supplementary Methods on how we capture sea level rise impacts on tides (lines 50-56). Ocean stratification effects on coastal flooding are small but known (e.g. applying methods of Orton et al. 2012 to Hurricane Sandy found a 1% effect on storm tide) and the effects of any small change in that stratification due to climate change, much smaller.

“Only limited locations with sparse populations meet this description, along parts of Greenland, Antarctica, and Alaska, mainly due to reduced local gravitational pull on the ocean because of diminished local land-ice mass...”

Mainly due to glacial isostatic adjustments (GIA). Check that!!! See e.g. Peltier

This paper is quantifying damage attribution for *anthropogenic global* sea level rise and not generally discussing *relative* sea level rise (see response to Reviewer 1, Question 2), but your point simply reminded us that these sentences are an unnecessary distraction. They have now simply been removed, for relative lack of importance to the main thrust of our paper.

“Most of this research assumes a simple 1:1 addition of sea level rise...”

Maybe „linear“ would be a better description

A good suggestion - we have made that change.

“...without any hydrodynamic modeling to capture known potential nonlinear effects (e.g., Orton et al, 2015; Arns et al, 2017).”

Well... not always known and still not 100% described by physics. I would delete the term „known“ or replace it by „observed“

Another good suggestion - we have changed the text to read “observed nonlinear effects”.

“None of it isolates the effect of the climate mediated human contribution to sea level rise from other factors such as natural variability and local land subsidence.”

At least none of the above mentioned... But there was a study (amongst others but i only remember this one) by Dangendorf et al. 2014 called evidence for long term memory in sea level discussing the anthropogenic effects on SLR.

Many papers discuss anthropogenic effects on SLR. We do not claim otherwise. The point we are making here is that no previous paper, to our knowledge, has isolated the effects of anthropogenic SLR on the damages from, or severity of, past coastal floods. Dangendorf et al.

2014, for example, do not discuss any specific past coastal floods. This is the core contribution of our manuscript, to isolate and estimate the effects of ASLR on damages from a historic flood. We have added the qualifier “to our knowledge” to the quoted sentence in the manuscript.

**"This study neglects any other possible effects of climate warming aside from sea-level rise (such as thermodynamic changes, potentially affecting storm track and intensity)."
...as well as changes in hydrodynamic properties (see e.g. Schindelegger et al., 2018 on changing M2 tides).**

Yes, another useful suggestion - we have addressed this point with a new paragraph (see fourth paragraph of Supplementary Methods, lines 50-56):

“Our hydrodynamic modeling captures the influence of sea level rise on tides within the model domain, including amplification within estuaries (e.g. Long Island Sound; Orton et al. 2019). However, it cannot capture any Atlantic basin-wide changes. At the open coast in the New York Bight region (e.g. Atlantic City, Sandy Hook) the total of these changes is below 1 cm per half-meter of sea level rise (Schindelegger et al. 2018), some fraction of which is captured in our model domain. Therefore, any missing basin-wide changes to tides due to sea level rise are on the order of a few millimeters, negligible for our study’s purposes.”

Also, in the second paragraph of that section, the large domain size is clarified: “ADCIRC is run on a pre-existing unstructured mesh covering the Northwest Atlantic, including the US East Coast and Gulf of Mexico, with 3.1 million nodes ...”

“Hurricane Sandy had a worst-case timing with respect to the evening high tide (Georgas et al, 2014) and a near-worst case storm track (Hall and Sobel, 2013), so any climate-related changes in the track and timing would likely have worsened damages.”

No! I don’t understand your statement (without describing it in more detail). Damages could also be less intense if the timing or tracks have changed.

We have made several changes to improve the clarity of this section (lines 55-64):

“Studies have so far found no evidence that Sandy’s intensity, size, or unusual storm track were made more likely by climate change (Lackmann, 2015; Mattingly et al., 2015; Orton et al. 2019). More broadly, a recent study found that future climate change effects on tropical cyclones will have only a small effect on extreme sea levels in New York Bight, relative to the effects of sea level rise (about 10% as much; Marsooli et al. 2019). Sandy had a worst-case timing with respect to the evening high tide (Georgas et al, 2014) and a near-worst case storm track (Hall and Sobel, 2013). Any differences from this scenario in a hypothetical world without climate change would likely have reduced damages. Conversely, if climate change did influence Sandy's actual track and timing, it would likely have worsened damages. We thus seek to quantify a lower bound for the effect of anthropogenic climate change through focusing on the influence of linked sea level rise alone.”

In other words, we are looking at past changes to climate, not future changes. Sandy's storm track and timing were essentially "worst-case", so if we were able to undo past climate changes to these factors, they could only ameliorate the flooding.

3 Flood modelling

"The ensemble of counterfactual runs includes mean sea level perturbations of -24, -20, -16, -14, -12, -10, -8, and -4 cm, relative to the historical run."

Where do these numbers come from? Maybe I did not get it from above but it definitely needs to be explained!

We have added explanation to the text: "This captures a broad range of possible uncertainty in ASLR; we sample more densely near the center of the distribution." (lines 346-347) The point is, we use estimates of damage and exposure at this limited set of levels (feasible to compute) to interpolate to the much larger number of levels that correspond to ASLR estimates using different methodologies and percentiles, etc.

4 Model validation and error reduction

"We exclude high water marks rated as \poor" quality or located in the wave-active FEMA V Zone..." Where rated? By whom? Which criteria?

Change made - we have added text referring to the US Geological Survey (USGS) reports giving these ratings.

5 Damage analysis

"Property damage is calculated using the using the..."

Delete the second "using the"

Done.

"Damages and exposures for precise amounts of ASLR (e.g., the 50th, 5th, and 95th percentiles of the pooled CMIP5-, Stable-, and Cooling-based scenarios) were interpolated between results for various water levels (from 4, 8, 10, 12, 14, 16, 20, and 24 cm of ASLR)." Until now I have not really understood what your ASLR runs are based on. You state above that it is the counterfraction runs assuming cooling and stable conditions. Where do 4-24 cm of ASLR come from? Maybe reword some above sections and try making it easier to follow for those not involved in the work.

The 4-24 cm of ASLR were chosen to span all the values between the 5th and 95th percentiles of the various scenarios so that we could interpolate to estimate damages for all of the percentile values considered. We could choose only a limited number of sea level perturbations to run because of computational costs. We sampled more heavily in the central part of the range, where more percentile values required estimates.

We feel that the edits made in response to your comment farther above, concerning the part of the text where we introduce our 4-24 cm perturbations, should sufficiently clarify the matter for most readers.

6 Results and discussion

“These errors were calculated using the ratio of percentage of damage attributed to ASLR under the S+D or S models in SI Table 2, respectively, relative to under the S+BC+D model (all based on the 50th percentile from CMIP5).”

No chance understanding this... what is D? What is BC?

These abbreviations have been eliminated from the main text, and are clearly explained in the caption for Supplementary Table 6, where they are useful abbreviations.

Reviewer #3 (Remarks to the Author):

The article sets out to quantify the contribution of anthropogenic sea level rise to the impacts of Sandy. The paper is overall well written and interesting, but at the moment I find that it has some major drawbacks which could even justify rejection for a high impact journal like NCOMMS. I explain better below:

The authors make clear that they focus on ASLR and not climate change in general. They consider previously published SLR projections, but one weakness of the study is that there is no mention on how climate change affected the probabilities to have an event like Sandy occur. I can understand that addressing this point may not be easy but in my opinion it is at least essential to discuss.

We have added text to clarify our decision, including:

“Studies have so far found no evidence that Sandy’s intensity, size, or unusual storm track were made more likely by climate change (Lackmann, 2015; Mattingly et al., 2015; Orton et al. 2019). More broadly, a recent study found that future climate change effects on tropical cyclones will have only a small effect on extreme sea levels in New York Bight, relative to the effects of sea level rise (about 10% as much; Marsooli et al. 2019).” (lines 55-59)

If I understand correctly the authors use a model which estimates global mean SLR. This implies the that MSL changes in New York follow the global mean. If this is the case is there any evidence to support this assumption?

We have substantially revised the manuscript to include New York-specific estimates of total absolute sea level rise and its attributable fraction.

I believe there is a lot of space for improvement in the description of the methods and data used. I include a lot of objections for the flood modeling in the paragraphs below.

Also the SLR projections have been published before but the current information is not sufficient to follow the work done. The reader is forced to read the PNAS paper to be able to obtain essential information needed to understand the current study.

We believe the descriptions included under “Semi-empirical attribution” (Results) and “Attribution from semi-empirical models” (Methods), plus associated figure and table captions, provide a sufficient summary of the previous work in the context of this manuscript, allowing readers to follow the main logic and details of the approach, all the way down to different calibrations of the semi-empirical model, and the pooling and sampling of results from different simulation sets.

But the weakest part of the work in my opinion is in the flood modeling.

We agree that the manuscript did not give adequate information on the modeling, and in this resubmission we add substantially more detail, mainly through new text added to the Supplementary Methods. It begins:

“The coupled ADCIRC/STWAVE model used here is one of the best available coastal flood models for a simulation over such a large area (the “Sandy-affected area”), balancing speed, resolution and achieving high accuracy. ADCIRC is a very widely-used model for coastal flood studies, including increasing adoption by the Corps of Engineers (e.g. the North Atlantic Coast Comprehensive Study) as well as by NOAA for real-time forecasting (https://ocean.weather.gov/estofs/estofs_surge_info.php) and post-disaster hurricane flood hindcasts (https://www.weather.gov/sti/coastalact_surgewg).” (lines 19-25)

The model mesh was created for the Corps of Engineers North Atlantic Coastal Comprehensive Study (NACCS), in response to Hurricane Sandy. The manuscript did refer to the extensive reports for the NACCS study (Cialone et al. 2015), but we have now added more of these details to the Supplementary Methods.

First of all the authors apply bias correction to the model (eq 1) which I find alienating. Error in a hydrodynamic model is normally a reason to do better calibration and is not something that can be solved with bias correction. The correction adds bias in the results of a process based model which is a problem when different scenarios need to be explored (as is presently the case). Given that the area has very good coverage of tide and wave gauges these could allow for a detailed validation/calibration of waves, water levels along the domain. Having hydrodynamic processes properly simulated by the model would be a precondition for accurate flood modelling. Then model errors could be dealt with calibration, more accurate surface roughness data, etc. All that background information is missing from the paper I think that the authors should show clearly that they did their best to oppress model errors before opting for bias correction.

There was a full model validation for the model prior to our study, and it captures the physics and the water levels well prior to the bias correction. As is increasingly the case with coastal hydrodynamic models, the model is accurate enough and captures the hydrodynamics of flooding well enough that “calibration” (tuning of poorly-constrained parameters) is no longer typically utilized (e.g. Thomas et al. 2019). The modeling methods are detailed in the NACCS reports (e.g. Cialone et al., 2015).

To put the Sandy simulation accuracy into perspective, we added text in the Supplementary Materials:

“The validation using pre-Sandy bathymetry (Supplementary Figure 1) shows RMS errors in maximum water elevation of 0.355 m on peak water elevations of up to 4.2 m. Therefore, the model is capturing the physical processes of storm surge and tide very well. For comparison, a 2019 model study of Hurricane Matthew’s coastal storm tides across several states had RMS errors on water level of 0.28 m, relative to a peak water elevation of 2.9 m

(Thomas et al. 2019).” (lines 57-62)

The Thomas paper in the journal Ocean Modelling (the top-rated journal in the field) gives excellent perspective on the accuracy of our simulation for a coastal flood simulation with the necessary high floodplain resolution yet large area required in our study.

We further note, “Spatially-coherent bias on the order of tens of centimeters (see Supplementary Figure 1, Pre-correction panel) is a common result with 2D ocean models that do not capture relatively subtle baroclinic effects such as buoyant coastal trapped flows (e.g. Orton et al. 2012), and methods for incorporating or correcting these errors in 2D models are an active area of research (e.g. Thomas et al 2019, Asher et al 2019). A possible other source for the bias includes erroneous flow through an inlet causing an inaccurate water level across an entire back bay. Further lowering the remaining random errors, which arise for high water marks in upland areas, is a challenge due to the constant evolution and required parameterization of flow over the built landscape (e.g. using a Manning's-n roughness on a mesh with a minimum resolution of 70 m).” (lines 75-84)

The text giving details on the bias correction was also moved to the Supplementary Methods, and the following summary text was added in the paper:

“...there is a clear spatial correlation to the error (Supplementary Figure 2, left panel). This is a common challenge with coastal flood modeling and can arise from baroclinic processes not captured in a 2D model (e.g. a buoyant coastal current) or erroneous (or storm-impacted) bathymetry in the inlet to a lagoonal back-bay estuary, several of which exist along the region's coasts. A spatially-coherent bias correction is utilized to correct for this component (see Supplementary Methods). The resulting bias-corrected modeled maximum water levels have a reduced mean bias... and the error field has no visible spatial correlation remaining (Supplementary Figure 2, right panel)... While correction does improve model accuracy, the effects on final results are small (see Supplementary Table 6).” (lines 357-366)

Similarly I don't understand why they chose to exclude tidal gauges inside wave action instead of pre-processing the time series to remove the wave effects (e.g. some low pass filter?).

We did not omit any tide gauges – we only omitted high-water marks (HWMs) in wave-affected areas, because it is impossible to filter out wave action for those cases. High-water marks are actual debris lines or other evidence of the temporal maximum water level. We have changed wording in the manuscript to avoid use of the wording “high water observations” to avoid confusion with HWMs, changing it to “maximum water level observations”.

It is mentioned that the modelled water levels are combined with LIDAR data. Does this mean that the LIDAR data are not included in the inundation simulations? Where they simulations

or the authors just applied a bathtub method. If there was simulation the resolution in land was 70 m as stated? If yes along the whole subaerial part of the study area? All these need to be better clarified in the methods.

The methods and supplementary sections have been updated to be much more thorough, answering these specific questions. Many quotes make clear that this is dynamic modeling, not bathtub methods – e.g. “Hydrodynamic modeling of sea-level-rise impacts on storm-driven flooding is generally justified because many prior studies have shown that there can be nonlinear interactions between coastal floods and sea level rise.” (lines 146-148) The best available DEM data, typically from LIDAR measurements, are utilized to create the DEM used by the model. However, hydrodynamic modeling always utilizes a mesh with a resolution that is far coarser than the source DEM data, which are averaged to the resolution of the mesh, which was typically about 70 m in upland areas flooded during Sandy. The mesh has a variable resolution that is maximal in nearshore areas and floodplains. A bathtub method was never used in this study, and we have edited the text to make sure we use the word “hydrodynamic” to refer to simulations/modeling in multiple places.

Once water surfaces were hydrodynamically modeled on the variable-resolution mesh, those surface elevations were compared against native ~5 m horizontal resolution land elevations to create the flood extent maps and water depth fields used for exposure estimates and damage modeling, as explained in the Damage analysis section (lines 368-373).

Also I am surprised that there is no validation of the model in terms of flood extents. As far as I know flood maps from Sandy are available.

There are no actual regional ‘observed’ flood extent maps for Sandy – what has been misrepresented in a few instances (e.g. Wang et al. 2014; Narayan et al. 2017) is the FEMA MOTF flood extent dataset, which is actually a spatial interpolation of observed maximum water level data (Georgas et al. 2014; FEMA 2013). Therefore, a comparison with the actual maximum water level data is the only true comparison with observations that is possible.

Regarding the damage analysis, again there are a lot of vague points. What is the resolution finally considered? Inundation depths are averaged for each census block, that should decrease the resolution and affect the accuracy of the estimates. One would expect a more detailed approach since a single event is studied and the spatial extent is rather limited. Another important limitation is that there is no consideration for indirect impacts which could imply a 50% or more underestimation of losses and distorts the results due to the fact that the relationship between direct and indirect losses is not linear. All the above are serious shortcomings and they are not even discussed while the methodology is very poorly described.

We agree with the reviewer that the amount of detail on the damage assessment in our original submission was limited. We therefore made several clarifying updates to the Damage analysis section (now under Methods, lines 367-397).

To address some of the specific points raised, we followed the same methodology as in Aerts et al. (2013) for the damage calculation. This is indeed done at the census block level, which is the standard unit used by HAZUS. We are unaware of affordable, nonproprietary data that would permit a more detailed analysis. However, we estimated the wet fraction of the census block in case it is not completely flooded. This percentage-flooded value was used to adjust the damage calculation of the entire census block. This approach improves the effective resolution of the analysis to a sub-block level, although it essentially assumes that within census blocks modeled, there is no correlation between elevation and value subject to damage.

The reviewer is correct to be concerned about indirect as well as direct damages. We have edited the text to address this point, adding new language to our Discussion:

“Whatever the value of ASLR, total economic damages may also be greater than damages indicated here. Our estimates do not account for potential long-term economic effects, in terms of losses and gains in broad economic activity associated with employment and production changes across industries in the aftermath of a damaging cyclone strike (Hsiang et al. 2014). Other studies of indirect damages find they may scale slightly exponentially with direct damages (e.g. exponent near 1.4, Koks et al. 2015), but may stay relatively low even for exceptional events (e.g. roughly one-third of more than \$100 billion in total damages for Hurricane Katrina (Hallegatte et al. 2008). Any nonlinear scaling would bias low our estimates of damages associated with ASLR, but this effect should be small, as our study concerns the difference between two very similar floods.” (lines 245-253)

Another issue relates to the figures which I don't find informative, especially given the fact that the article has been submitted to a high impact factor journal. Figures 2 and 3 could improve aesthetically but most important seem to be more methodological and thus more appropriate for the SI. There are no spatial maps comparing flood extents, and most importantly the distribution of damages and how these vary among the compared scenarios. Apart from the tables and Fig 1 there is no display of uncertainty and confidence levels. It seems to me that the authors did not put the amount of work needed to the manuscript.

Helpful points. We have moved Figures 2 and 3 from the original manuscript to Supplementary Figures. We have added a new figure (new Figure 3) mapping, in panel a, the difference between modeled flood extents under three different counterfactual scenarios (essentially spanning the 5th- to 95th-percentile range across all ASLR models) versus the flood as it occurred. Panel b of the figure shows the modeled flood depth differences between the actual flood and a scenario with 10 cm of ASLR (close to the total ensemble ASLR estimate), clearly illustrating the much greater prevalence of depth differences.

Another new figure (new Figure 2) maps the spatial distribution of damage differences (from actual) under the same three counterfactual scenarios. And the new Supplementary Figure 3 shows 5th-95th percentile uncertainty ranges of damage differences by county.

Finally, I found the literature review outdated. The authors cite one paper from 2017 and one from 2018, while during these past years there have been several interesting contributions related to coastal hazards and impacts including many in high impact journals (addressing future changes in climate extremes, developments in coastal flood modelling and their implications for coastal losses).

More than a dozen references from 2017 or more recent have been added addressing these and other areas, including Dangendorf et al (2017), Marcos et al (2017), Marsooli et al. (2019), Oppenheimer et al. (2019), Orton et al. (2019), and Valle-Levinson et al (2017).

REVIEWER COMMENTS

Reviewer #1 (Remarks to the Author):

I thank the authors for their careful revision of the paper and response to my comments. The authors have satisfactorily addressed my specific concerns. In reading through the manuscript the first time, I did have concerns about the flood modeling, but recognized this is outside my area of expertise. In reading the other reviews and associated response, I would not advocate for publication unless the concerns regarding the flood modeling raised by the third reviewer are met. It's a critical component of the paper and the conclusions therein. I leave that up to the editor and other reviewer, however.

Reviewer #2 (Remarks to the Author):

The authors took into account my previous comments, and the manuscript has been significantly improved. I recommend this manuscript for publication, but would like to suggest to the authors to take one last comment into account:

Fig.3: Creating nice figures is always tricky and a matter of taste. However, one they should always be self-explaining and the required information always needs to be accessible without having to read the entire manuscript. In its current form, I have problems to understand/identify the content you refer to in a). Blue areas are only visible when zooming in and it is hardly possible to see anything in a printed version. Also, when using colours, I personally always try to include the colourbar/range that has been used. I would suggest to spent another round of revisions on the figure in order to increase its relevance.

Reviewer #3 (Remarks to the Author):

I acknowledge that the authors improved significantly the paper and responded to most comments. While the first version was substandard in several aspects the current one is well written and presented. However, I am afraid that I cannot recommend the paper for publication in NCOMMS and the reason is the scientific question per se. The study focuses on the additional flooding and economic impacts of Sandy due to anthropogenic climate change (ACC). I have two main objections: Weather is such a complex system and there are so many unique stochastic conditions that made Sandy the event it was. So on the physical domain and despite the justifications, I am not fully convinced that Sandy would have been the same event without the historical ACC. Let's keep in mind that 0.1 degree change in the orientation and few hours lag in the landfall can make a huge difference!

But this is not my main objection, as my concern is about the economic damages which are served as the 'main course'. Carbon consumption has been the main driver of the industrial revolution and the economic boom our societies experienced recently. I understand that one can estimate the marginal flooding without ACC, but the latter has affected socioeconomic development so much that without taking it in account, any comparison of economic damages is meaningless. ACC has been the side effect of socio-economic conditions which made NY the city it is now. So the cost of flooding without ACC would need to be based on what would be the exposure, vulnerability and asset values in the scenario that we didn't have carbon-related growth. The non-ACC NY would be such much poorer and smaller that the economic damages would be orders of magnitude lower, so the marginal difference would be much higher. Still I am suspecting that the damages as a fraction of the GDP would be higher in the non-ACC world. Even though that would be an interesting experiment I doubt it would have a real-life meaning.

So my recommendation would be to remove the calculation of the economic costs and try submitting to a lower impact-factor journal.

Reviewer #4 (Remarks to the Author):

Nature journals are a pain to review because the paper is just a glossy announcement and all the details are hidden in appendix and references.

In this case, the details are really hidden. The paper mentions a "damage model" once and "damage modeling" thrice, but no description of said model can be found anywhere.

It may be that the authors forgot to describe the model, in which case the paper should be completed and resubmitted for review.

It may be that the authors assumed that damage is proportional to flood depth or something, in which case they should go do their homework.

Reviewer #5 (Remarks to the Author):

Thanks for the opportunity to review this excellent paper. I found it very well done, extremely clear and well written with due disclosure of the limits of the study. The topic is important, both scientifically and practically, and the paper uses a very solid methodology.

My only comment or question is the role of flood defenses. In modeling work on coastal floods (for instance, Hallegatte et al. 2013), average annual flood losses increase very rapidly with changes in average sea level, much more than what is suggested here. The rationale is that cities have protection against floods that act up to a certain local water level that is reached rare (every 100 years in many places, up to thousands of years in some places). When these defenses are overtopped or fail however, losses become quickly very large (think of New Orleans). A limited change in average sea level can change drastically the frequency of water levels above the defense levels.

Here, the problem is different before we look at a single event (which I find acceptable, I do not see why one reviewer finds that problematic), but I'm still unsure how flood defenses are accounted for. If manmade flood defenses are absent from the topographical information, which seems likely based on the data used and the authors' response to other referees, then the analysis may miss some threshold effects.

Since Sandy had a very high water level, defenses were likely overtopped, so it's possible that the hydrological modeling does not need to account for defenses to reproduce the footprint of the Sandy flood. But if the scenario without ASLR makes the water level be just lower than the defenses levels, then losses could drop to zero (basically the flood disappears).

Calibration of the flood footprint and economic losses on the Sandy case does not tell us much about this problem because the defenses are likely overtopped. To confirm that this problem does not affect results, the flood footprint model should be validated against other, lower, water level.

If that factor plays a role, then the fraction of the losses attributed to sea level rise may be strongly underestimated. Actually, if water level exceeds defense levels only with ASLR, then 100% of losses are due to ASLR. I'm not saying it's the case, only suggesting that the author(s) discuss(es) this issue.

This limit (if it is real, as I may simply have misunderstood the methodology) would need to be discussed, but the paper remains a very valuable contribution.

Hallegatte, S., Green, C., Nicholls, R. J., & Corfee-Morlot, J. (2013). Future flood losses in major

coastal cities. *Nature climate change*, 3(9), 802-806.

Stephane Hallegatte

REVIEWER COMMENTS

Reviewer #1 (Remarks to the Author):

I thank the authors for their careful revision of the paper and response to my comments. The authors have satisfactorily addressed my specific concerns. In reading through the manuscript the first time, I did have concerns about the flood modeling, but recognized this is outside my area of expertise. In reading the other reviews and associated response, I would not advocate for publication unless the concerns regarding the flood modeling raised by the third reviewer are met. It's a critical component of the paper and the conclusions therein. I leave that up to the editor and other reviewer, however.

Reviewer #2 (Remarks to the Author):

The authors took into account my previous comments, and the manuscript has been significantly improved. I recommend this manuscript for publication, but would like to suggest to the authors to take one last comment into account:

Fig.3: Creating nice figures is always tricky and a matter of taste. However, one they should always be self-explaining and the required information always needs to be accessible without having to read the entire manuscript. In its current form, I have problems to understand/identify the content you refer to in a). Blue areas are only visible when zooming in and it is hardly possible to see anything in a printed version. Also, when using colours, I personally always try to include the colourbar/range that has been used. I would suggest to spent another round of revisions on the figure in order to increase its relevance.

>> We have added a color legend to Fig 3a and edited Fig 3a labeling to improve clarity. We recognize it is difficult to see the blue areas, but believe that helps to communicate results through its contrast with the larger orange and red zones. We could create a much further zoomed-in figure that would make it easier to see narrow blue margins, but in our opinion the tradeoff of having a much smaller geographic field of view outweighs being able to see more of the blue layer.

Reviewer #3 (Remarks to the Author):

I acknowledge that the authors improved significantly the paper and responded to most comments. While the first version was substandard in several aspects the current one is well written and presented. However, I am afraid that I cannot recommend the paper for publication in NCOMMS and the reason is the scientific question per se. The study focuses on the additional flooding and economic impacts of Sandy due to anthropogenic climate change (ACC).

I have two main objections:

Weather is such a complex system and there are so many unique stochastic conditions that made Sandy the event it was. So on the physical domain and despite the justifications, I am not fully convinced that Sandy would have been the same event without the historical ACC. Let's keep in mind that 0.1 degree change in the orientation and few hours lag in the landfall can make a huge difference!

>> We agree that Sandy may have been a different storm without historical ACC. The manuscript acknowledges this possibility, but clearly and explicitly isolates the impact of anthropogenic sea level rise on Sandy's flood as its focus, which we believe is a legitimate and important exercise on its own.

But this is not my main objection, as my concern is about the economic damages which are served as the 'main course'. Carbon consumption has been the main driver of the industrial revolution and the economic boom our societies experienced recently. I understand that one can estimate the marginal flooding without ACC, but the latter has affected socioeconomic development so much that without taking it in account, any comparison of economic damages is

meaningless. ACC has been the side effect of socio-economic conditions which made NY the city it is now. So the cost of flooding without ACC would need to be based on what would be the exposure, vulnerability and asset values in the scenario that we didn't have carbon-related growth. The non-ACC NY would be such much poorer and smaller that the economic damages would be orders of magnitude lower, so the marginal difference would be much higher. Still I am suspecting that the damages as a fraction of the GDP would be higher in the non-ACC world. Even though that would be an interesting experiment I doubt it would have a real-life meaning.

So my recommendation would be to remove the calculation of the economic costs and try submitting to a lower impact-factor journal.

>>Reviewer #3 provided many insightful comments in the first round of review which helped us to greatly strengthen the manuscript, and for which we are grateful. Here, the reviewer makes a broad and basic philosophical argument (equally applicable to our original manuscript, but missing from the reviewer's first set of comments) which, if correct, would delegitimize attribution to anthropogenic climate change of damages from any current or past event whatsoever. We disagree strongly with this point of view, and further feel that to address it in the manuscript would be a detour and distraction not appropriate for a more focused study such as our own. (It could make a topic for a commentary.) Our purpose was not to create a truly complete counterfactual scenario, such as accounting for all the differences in the storm that might have been, let alone developing an alternative history of global economic development (an impossible and unverifiable exercise). Rather, we seek to isolate the effect of just one factor --anthropogenic SLR -- for controlled study, following longstanding and basic scientific methodology. Our object is not to imagine how Sandy would have unfolded without anthropogenic climate change (and the economic development which caused it), but rather to better understand the costs of anthropogenic climate change, and thus to contribute to a broader discussion of costs and benefits. The economic development that has led to anthropogenic warming has created many benefits, and it has caused many costs. To estimate one of the specific costs, as we do here, it is not necessary to construct a holistic counterfactual world -- in fact, it would be counterproductive.

If the editors still insist that we address this comment, we propose inserting the following paragraph just before the final one of the discussion:

“A complete counterfactual history without anthropogenic climate change might include slower 20th-century economic development, leading to lower counterfactual damages from Sandy -- and a greater difference vs. actual -- because less economic value would exist to be damaged. This paper instead develops the more feasible and controlled thought experiment of isolating the effect of one variable alone, anthropogenic SLR.”

Reviewer #4 (Remarks to the Author):

Nature journals are a pain to review because the paper is just a glossy announcement and all the details are hidden in appendix and references.

In this case, the details are really hidden. The paper mentions a "damage model" once and "damage modeling" thrice, but no description of said model can be found anywhere.

It may be that the authors forgot to describe the model, in which case the paper should be completed and resubmitted for review.

It may be that the authors assumed that damage is proportional to flood depth or something, in which case they should go do their homework.

>> We have added a more detailed description of the damage model by substantially updating the third paragraph of section 4.4 (in the methods) to the following (new text italicized):

Property damage is calculated using the standard HAZUS-MH method from the U.S. Federal Emergency Management Agency (Scawthorn 2006), *using the same approach as utilized in Aerts et al (2014), which is described in detail in their supplementary information. In short, the damage calculation is based on depth-damage curves, which denote the vulnerability of a specific asset to different flood depths in the form of a percentage of maximum flood damage (which is based on depreciated replacement costs). The curves are typically non-linear, rising sharply through low inundation depths and levelling off with higher ones, and are based on observational studies using insurance claims data from the National Flood Insurance Program (FEMA 2020). Both depth-damage curves and maximum damages are differentiated for 33 building types ranging from various residential types (single home, multi-story apartment buildings, etc.) to, amongst others, industrial, commercial and educational buildings. Counts of these building types are available per census block from the HAZUS databases. Therefore, we compute mean inundation depth for flooded areas within each Census block, use this depth to compute hypothetical full-block damage for all buildings present in the Census block, and reduce this total by the fraction of area not flooded. We assume that all buildings (and types) are distributed equally within the Census block.*

Reviewer #5 (Remarks to the Author):

Thanks for the opportunity to review this excellent paper. I found it very well done, extremely clear and well written with due disclosure of the limits of the study. The topic is important, both scientifically and practically, and the paper uses a very solid methodology.

My only comment or question is the role of flood defenses. In modeling work on coastal floods (for instance, Hallegatte et al. 2013), average annual flood losses increase very rapidly with changes in average sea level, much more than what is suggested here. The rationale is that cities have protection against floods that act up to a certain local water level that is reached rare (every 100 years in many places, up to thousands of years in some places). When these

defenses are overtopped or fail however, losses become quickly very large (think of New Orleans). A limited change in average sea level can change drastically the frequency of water levels above the defense levels.

Here, the problem is different before we look at a single event (which I find acceptable, I do not see why one reviewer finds that problematic), but I'm still unsure how flood defenses are accounted for. If manmade flood defenses are absent from the topographical information, which seems likely based on the data used and the authors' response to other referees, then the analysis may miss some threshold effects.

Since Sandy had a very high water level, defenses were likely overtopped, so it's possible that the hydrological modeling does not need to account for defenses to reproduce the footprint of the Sandy flood. But if the scenario without ASLR makes the water level be just lower than the defenses levels, then losses could drop to zero (basically the flood disappears).

Calibration of the flood footprint and economic losses on the Sandy case does not tell us much about this problem because the defenses are likely overtopped. To confirm that this problem does not affect results, the flood footprint model should be validated against other, lower, water level.

If that factor plays a role, then the fraction of the losses attributed to sea level rise may be strongly underestimated. Actually, if water level exceeds defense levels only with ASLR, then 100% of losses are due to ASLR. I'm not saying it's the case, only suggesting that the author(s) discuss(es) this issue.

This limit (if it is real, as I may simply have misunderstood the methodology) would need to be discussed, but the paper remains a very valuable contribution.

Hallegatte, S., Green, C., Nicholls, R. J., & Corfee-Morlot, J. (2013). Future flood losses in major coastal cities. *Nature climate change*, 3(9), 802-806.

Stephane Hallegatte

>>We agree it is important with any coastal hydrodynamic modeling that the grid/mesh must resolve defenses to capture threshold flooding processes. Below, we address why our modeling satisfies this demand. Text was also added to point out this important issue, modifying and better organizing some text that was unclear in the Supplementary Methods section (second paragraph):

“An important concern with hydrodynamic flood modeling is adequate resolution to capture any flood defenses, which can be linear features of small scale. The ADCIRC mesh used in this study is the same 3.1-million-node mesh used in the North Atlantic Coast Comprehensive Study (NACCS) (Cialone et al, 2015). The portion of the NACCS mesh covering the New York area was expanded, enhanced and updated from a FEMA Region II Flood Risk Map Study (FEMA,

2014), where the mesh was specifically designed to have nodes aligning with coastlines and river boundaries and vertical features such as boardwalks, jetties, roadways as well as existing flood defenses such as dunes, berms and floodwalls. In the Sandy-affected areas the mesh resolution was typically 70 m. Elevated vertical features such as shoreline berms have resolutions as low as 50 m and were delineated in mesh creation so as to optimally resolve the elevated spine of the feature. Small-scale geographical features such as tributaries have a resolution as low as 10 m (Cialone et al, 2015). The mesh underwent extensive validation as part of the North Atlantic Coast Comprehensive Study (Cialone et al, 2015) including extratropical storm events that did not produce as much flooding as Sandy. ”

It is a reasonable general concern with any coastal flood modeling that the grid/mesh must resolve defenses to capture threshold flooding processes, but it is standard practice to handle this with models of a high resolution or linear features between cells such as weirs or thin dams. In the case of our modeling of Hurricane Sandy, subtracting off a sea level rise increment of up to 24cm, threshold effects should be very rare and also are generally captured with our hydrodynamic model. This is for the following reasons:

- The geography of New York City and the broader region is not deltaic, and all populated areas are well above normal high tide levels. For most locations, when flooding occurs, it doesn't pour into a lower protected basin like it can in places like New Orleans or The Netherlands. An exception is the berm-protected neighborhood of Midland Beach, Staten Island, but the ~250m wide roadway berm there is well-resolved by the model and was overtopped by about one meter (**Figure R1**).
- The model's DEM generally captures any flood defenses that exist that are of a sufficient scale to be incorporated into the model's DEM which has resolution as low as 50 m around linear shoreline features like beach dunes and land berms. Also, these types of linear features were specifically delineated so as to optimally resolve their elevated spine.
- Most urbanized areas of the region had no flood defenses, or they were very low relative to Sandy's peak water levels. (As Hallegatte et al. 2013 noted, as part of a global comparison, New York City has a “low protection level” and is among the global cities very most at risk of coastal flooding damages.) Most areas had a waterfront that was not raised above the neighborhood behind it. While the peak water level during Sandy was 3.4 m NAVD88 (at The Battery gauge, Manhattan), the peak observed water level there in the prior 190 years was only 2.2 m (Orton et al. 2016), making no observation-based validation possible between these levels, including at just below Sandy's level, as would be ideal. However, during Sandy, almost all waterfronts were overtopped by a significant margin, far greater than the small sea level increments we are studying. An example graphic that shows simplified flood thresholds and the wide margin by which Sandy exceeded flood thresholds is given in **Figure R2** (from Orton et al., 2019).

Finally, we note that we did identify one small area within the study region where the flood overcame coastal protections but where it appears the protections would have functioned substantially more effectively if a 95th-percentile estimate of anthropogenic SLR were deducted from the base sea level, according to our modeling: Keansburg, NJ, as noted in Supplementary Figure 3. The damages to Monmouth County, which encompasses Keansburg, represent 7

percent of modeled tristate damages as Sandy occurred; about 8 percent in the median ASLR case; and about 6 percent in the 95th-percentile case, exerting little effect on the overall result. Keansburg is protected on its waterfront by a sand dune system of variable height, as captured by our mesh, which was overtopped in some locations during Sandy's actual flood, as captured by our modeling. Levees extend inland, and flood waters were additionally able to make their way around the levees in our modeling. We have made minor updates to the caption to Supplementary Figure 3 to further clarify this point.

Figure R1 : A rare case in the region where a neighborhood is protected by a berm is Midland Beach, Staten Island, yet the model resolves the topography well. Peak water levels there during Sandy were 3.9 m, overtopping the berm by ~1 m.

Figure R2 : (Note that Imperial units were used here for a prior New York City related publication). Vertical scale bar illustrating approximate breach elevations for the water level (in feet) that floods various New York City locations and neighborhoods. Hurricanes Sandy and Donna peak water levels are shown for comparison. Water levels are assumed spatially constant. Breach, or critical, elevations estimated using a 1-foot horizontal resolution 2010 LIDAR-based DEM, with static mapping and 0.5-ft vertical increments of water level. Figure taken from Orton et al. (2019).

References

FEMA. (2014). Region II Storm Surge Project – Mesh Development. Federal Emergency Management Agency, US Department of Homeland Security, Washington DC.

Orton, P. M., T. M. Hall, S. Talke, A. F. Blumberg, N. Georgas, and S. Vinogradov (2016), A Validated Tropical-Extratropical Flood Hazard Assessment for New York Harbor, *J. Geophys. Res.*, 121, doi:10.1002/2016JC011679.

Orton, P., N. Lin, V. Gornitz, B. Colle, J. Booth, K. Feng, M. Buchanan, and M. Oppenheimer (2019), New York City Panel on Climate Change 2019 Report Chapter 4: Coastal Flooding, *Ann. N. Y. Acad. Sci.*, 1439, 95-114, doi:10.1111/nyas.14011.

REVIEWERS' COMMENTS

Reviewer #4 (Remarks to the Author):

It is unfortunate that key parts of the model are only disclosed after a referee asked for it.

The paper refers to Aerts et al. (2014), but methodologically this is Schneider and Chen (1980), what Steve used to call you the coloring book approach. The physics are on par with Minecraft.

There is no coastal protection, no behavioural response, no water flowing, no debris floating ... nothing but a digital elevation model and impact curves.

At the very least, the authors should come clean just how limited and simple their approach is. The authors have a dilemma here: The model is a local approximation at best, and they therefore cannot claim that climate change had a big effect on Sandy's damages; the approximation would simple be no longer valid.

But maybe it would be better to stop using this method.

Reviewer #5 (Remarks to the Author):

I'm happy with the detailed answers provided by the authors (and thanks for pointing me toward information already present in the S.I., apologies for missing it).

Reviewer #4 (Remarks to the Author):

Responses follow >>

It is unfortunate that key parts of the model are only disclosed after a referee asked for it.

The paper refers to Aerts et al. (2014), but methodologically this is Schneider and Chen (1980), what Steve used to call you the coloring book approach. The physics are on par with Minecraft.

>> We understand that the reviewer is referring specifically to the economic damage modeling. HAZUS-MH is based on observed empirical relationships between flood depth and damage, and does not take the approach of applying a physical model to each event. This was already addressed in previous versions of the manuscript, but new additions to section 4.4 (highlighted below) further indicate that we do not employ a mechanistic approach for damage modeling.

There is no coastal protection, no behavioural response, no water flowing, no debris floating ... nothing but a digital elevation model and impact curves.

>> Coastal protection is not a major factor in the Sandy-affected area. Furthermore, our modeling accounts for the protection that existed at the time of storm landfall, as noted in the Supplementary Methods and in our previous response to Reviewer #5. With respect to the other factors noted, we have added language to section 4.4 to caveat that our approach does not explicitly account for most of them:

“This approach does not explicitly address variability in non-depth mechanisms of damage that may include floating debris, chemical contamination, erosion, behavioral responses and water speed (except that it does use adjusted curves in strong wave action zones).”

Because HAZUS-MH depth-damage curves are based on a large number of empirical observations of events where a wide range of factors were in play, the mechanisms listed are implicitly addressed to some degree.

At the very least, the authors should come clean just how limited and simple their approach is. The authors have a dilemma here: The model is a local approximation at best, and they therefore cannot claim that climate change had a big effect on Sandy's damages; the approximation would simply be no longer valid.

But maybe it would be better to stop using this method.

>> As underscored in section 4.4, we believe that using HAZUS-MH as we do is appropriate for the objectives of this study, which focuses on differences in damages related to depth:

“meticulous absolute damage estimates are not essential to our analysis; rather, we aim to estimate relative damages among simulated floods starting from slightly different baseline sea levels, and this goal aligns well with the emphasis on depth in the HAZUS-MH model.”

More broadly, section 4.4 makes clear that our damage-modeling approach is based on a widely-used tool that employs empirical relationships and not detailed or mechanistic physical modeling.